# *Plasmodium falciparum* expresses fewer *var* genes at lower levels during asymptomatic dry season infections than clinical malaria cases

Sukai Ceesay[1,2☯], Martin Kampmann[1,3☯], Lasse Votborg-Novél[1,3],
Helle Smedegaard Hansson[4], Rasmus Weisel Jensen[4], Manuela Carrasquilla[1],
Hamidou Cisse[5], Louise Turner[4], Usama Dabbas[1], Christina Ntalla[1], Silke Bandermann[1],
Safiatou Doumbo[5], Didier Doumtabe[5], Aissata Ongoiba[5], Kassoum Kayentao[5],
Boubacar Traore[5], Peter D. Crompton[6], Thomas Lavstsen[4], Silvia Portugal[1]*

1 Max-Planck-Institute for Infection Biology, Berlin, Germany, 2 Charité – Universitätsmedizin Berlin, Berlin,Germany, 3 Humboldt-Universität zu Berlin, Berlin, Germany, 4 Centre for Translational Medicine and Parasitology, Department of Immunology and Microbiology, University of Copenhagen and Department of Infectious Diseases, Righospitalet, Copenhagen, Denmark, 5 Mali International Centre of Excellence in Research, University of Sciences, Techniques and Technologies of Bamako, Bamako, Mali, 6 Laboratory of Immunogenetics, National Institute of Allergy and Infectious Diseases, Division of Intramural Research, National Institutes of Health, Rockville, Maryland, United States of America

☯ These authors contributed equally to this work.
* portugal@mpiib-berlin.mpg.de

## Abstract

In seasonal transmission areas, clinical malaria occurs during the wet season when mosquitoes are present, while in the dry season, malaria transmission is interrupted and clinical cases are rare. In Mali, *Plasmodium falciparum* can persist in low parasitaemic asymptomatic individuals through the six-month dry season and shows circulation of more developed parasite stages compared to clinical malaria cases, indicative of reduced cytoadhesion of infected erythrocytes. How prolonged circulation of infected erythrocytes is achieved remains unknown. Here, we explored *var* gene expression in subclinical infections and clinical malaria cases of Malian children, collected during the dry and wet seasons. We sequenced expressed *var* DBLα-tags, used bioinformatic tools to predict their domain composition, binding phenotype and upstream sequence type; and determined their relationship to seasonality and clinical presentation. We found that parasites of asymptomatic infections expressed fewer *var* genes, with a larger proportion of *var* transcripts attributed to one or a few *var*s. In contrast, clinical cases exhibited expression of many *var* genes at lower proportions. We found that parasites of asymptomatic carriers expressed a mixture of CD36- and EPCR-binding PfEMP1, which changed over time. We confirmed that *vars* encoding CD36-binding PfEMP1 dominated in non-severe malaria cases, and found no significant difference in expressed *var* types between dry and wet seasons. Asymptomatic carriers were older, had higher titers of anti-*P. falciparum* antibodies, and broader reactivity to PfEMP1, suggesting that host immunity was the

**Data availability statement:** Data generated is available in the Edmond repository, is accessible under the following URL: https://edmond.mpg.de/dataset.xhtml?persistentId=-doi:10.17617/3.MUIYC0&faces-redirect=true

**Funding:** This work was supported by the European Research Council under the European Union Horizon 2020 Research and Innovation Programme (grant agreement 759534) (SP) and the Lise Meitner Excellence Programme of the Max Planck Society (SP). The Mali cohort study was funded by the Division of Intramural Research, National Institute of Allergy and Infectious Diseases, National Institutes of Health (PDC). The funders had no role in study design, data collection and analysis, decision to publish, or preparation of the manuscript.

**Competing interests:** The authors have declared that no competing interests exist.

main determinant limiting *var* transcript variation in asymptomatic carriers. However, qRT-PCR analyses also indicated higher total *var* transcript levels in malaria cases compared to asymptomatic carriers, suggesting that in addition to the parasite's switching and the host's immune selection of expressed *var* genes, parasites able to sustain long-term infections may be poised for reduced PfEMP1 expression.

## Author summary

In regions like Mali, where malaria is seasonal, *Plasmodium falciparum* can persist in asymptomatic individuals during dry months when mosquito-borne transmission is interrupted. The mechanisms enabling parasite survival in asymptomatic hosts throughout the dry season remain unclear. Here, we investigated expression of the *var* gene family, a key determinant of infected erythrocyte cytoadhesion, throughout the year. Our analyses of DBLα-tag sequences revealed fewer *var* genes expressed in asymptomatic infections, at higher relative proportions; whereas clinical malaria cases exhibited broader *var* gene expression at lower individual proportions. We found no significant seasonal differences in expressed *var* types of *P. falciparum*, and could detect *var* genes switching over time in asymptomatic infections. Asymptomatic carriers were older and had higher anti-*P. falciparum* antibody titters, supporting that host immunity limits *var* transcript diversity. Additionally, we observed significantly lower total *var* transcript levels in asymptomatic infections compared to clinical cases, suggesting that both host immunity and parasite-driven modulation of *var* gene expression contribute to the persistence of *P. falciparum* during the dry season.

## Introduction

*Plasmodium falciparum* remains a major global health problem, leading to over 300 million malaria cases, and 500.000 deaths every year, especially in young African children [1]. To evade host immunity and potentiate persistent infections, parasites possess several multicopy gene families encoding variant surface antigens (VSAs), of which the *P. falciparum* erythrocyte membrane protein 1 (PfEMP1) are particularly important as they mediate the cytoadhesion of the late stage infected red blood cells (iRBC) to specific receptors on the host endothelium [2–4]. Sequestration of iRBCs allows the parasite to avoid the splenic clearance of rigid and deformed iRBCs [5–7], and thus potentiates parasitaemia as well as malaria pathogenesis through blood occlusion and local inflammatory responses to iRBCs [8–15]. Expression on the iRBC surface makes PfEMP1 an immune target and entails switching of expressed variants to evade immunity and prolong infection [16]. Each parasite encodes ~60 *var* genes that are monoallelically expressed [17]. The overall repertoire is extensive [18,19], with new gene variants shuffled through meiotic recombination in the mosquito, and acquisition of mutations and mitotic recombination within the human host [20–26].

Nevertheless, each parasite repertoire is composed of similar *var* gene types, conferring similar repertoires of PfEMP1 phenotypes.

*var* genes consist of a highly variable Exon 1, encoding the protein's N-terminal extracellular part and the transmembrane domain, and a shorter more conserved Exon 2 encoding the protein's intracellular portion [27]. Most PfEMP1 contain an N-terminal head structure with a Duffy-binding-like (DBL)α domain followed by a Cysteine-rich-interdomain-region (CIDR) domain, followed by 2–6 C-terminal DBL and CIDR domains, a transmembrane domain, and an intracellular acidic terminal segment (ATS) at the C-terminus [27]. Based on chromosomal arrangement and upstream sequence (UPS), *var*s are classified into UPS groups A, B, C, and E [28,29], with each genome typically containing ~10% UPSA, ~75% UPSB and C genes [18] and one UPSE gene also known as *var2csa*. This grouping also reflects the clinical association and the primary receptor binding phenotype of the encoded PfEMP1, determined by the protein's N-terminal domain composition. Thus, *var2csa* PfEMP1 binds placental chondroitin sulphate A and is associated with placental malaria [30,31]. Severe malaria outcomes are linked to PfEMP1 binding endothelial protein C receptor (EPCR) via their CIDRα1 domains [12,32–34], which are encoded by approximately half of the UPSA *var* genes and a small subset of UPSB *var* genes, which likely arose from UPSB/A recombination events. Most UPSB and UPSC *var* genes encode PfEMP1s that bind CD36 through their CIDRα2–6 domains, and have been linked to uncomplicated malaria [35–39]. Expression of *var* genes in *P. falciparum* asymptomatic infections is scarcely studied. A few studies suggest a potential association of in asymptomatic infections with UPSC *var* genes [13,40], or report a reduction of UPSA and B but not UPSC in asymptomatic infections vs clinical malaria [41], suggesting a link between *var* gene regulation and asymptomatic infections. Others associate malaria immunity with expression of a restricted *var* repertoire in asymptomatic infection [42], although *var* gene switching appears to continue in longitudinally monitored samples [43]. The ordered nature of PfEMP1 phenotypes and corresponding acquisition of antibodies underlie the quick acquisition of immunity after few severe malaria or pregnancy malaria cases [44,45]. This highlights that host immunity inhibits the growth of parasites expressing certain *var*s, and raises the question of whether, and how, the parasite's *var* repertoire is utilized to facilitate asymptomatic infections bridging wet seasons several months apart.

In areas where malaria is seasonal, transmission is interrupted when mosquito breeding sites are unavailable due to lack of water; and clinical cases are rare [8,46–48]. During a months-long period, asymptomatic infections form the reservoir that restarts transmission in the ensuing wet season [48–50]. These infections exhibit extended circulation of iRBCs compared to those of clinical cases, evidenced by the presence of late ring and trophozoite stages in circulation [8], that is suggestive of reduced cytoadhesion. How the parasite addresses the challenge of persisting silently during periods free of mosquitos, until seasonal rains allow renewed transmission is not known. Here, to interrogate *var* gene expression in subclinical infections and clinical malaria cases of Malians collected during the dry and wet seasons, respectively, we sequenced RT-PCR-amplified *var* cDNA of 82 samples collected from 67 individuals throughout the year.

## Results

### Expressed DBLα-tag sequencing in a longitudinal cohort in Kalifabougou, Mali

In Kalifabougou, an area of Mali with intense seasonal malaria transmission, we investigated *P. falciparum* parasites from individuals enrolled in a previously described cohort study comprising ~600 individuals from 3 months to 45 years of age [51]. As typical for the region, in 2019, clinical episodes of malaria (≥2,500 asexual parasites per µl of blood and axillary temperature ≥37.5 °C) were frequent during the wet season (June–December), whereas nearly all individuals remained free of malaria symptoms during the dry season (January–May) (Fig 1A). The prevalence of *P. falciparum* asymptomatic infection was cross-sectionally examined in January, at the beginning (startDry), and in May, at the end (endDry) of the dry season by PCR [52], finding 18% and 10% of positive individuals, respectively. In October (midWet), during the ensuing wet season, 36% of cross-sectionally examined individuals tested positive (Fig 1B), in line with previous descriptions [8,48]. In the cohort, individuals presenting with clinical malaria (MAL) were younger (median 9 years, IQR 3 – 14) than

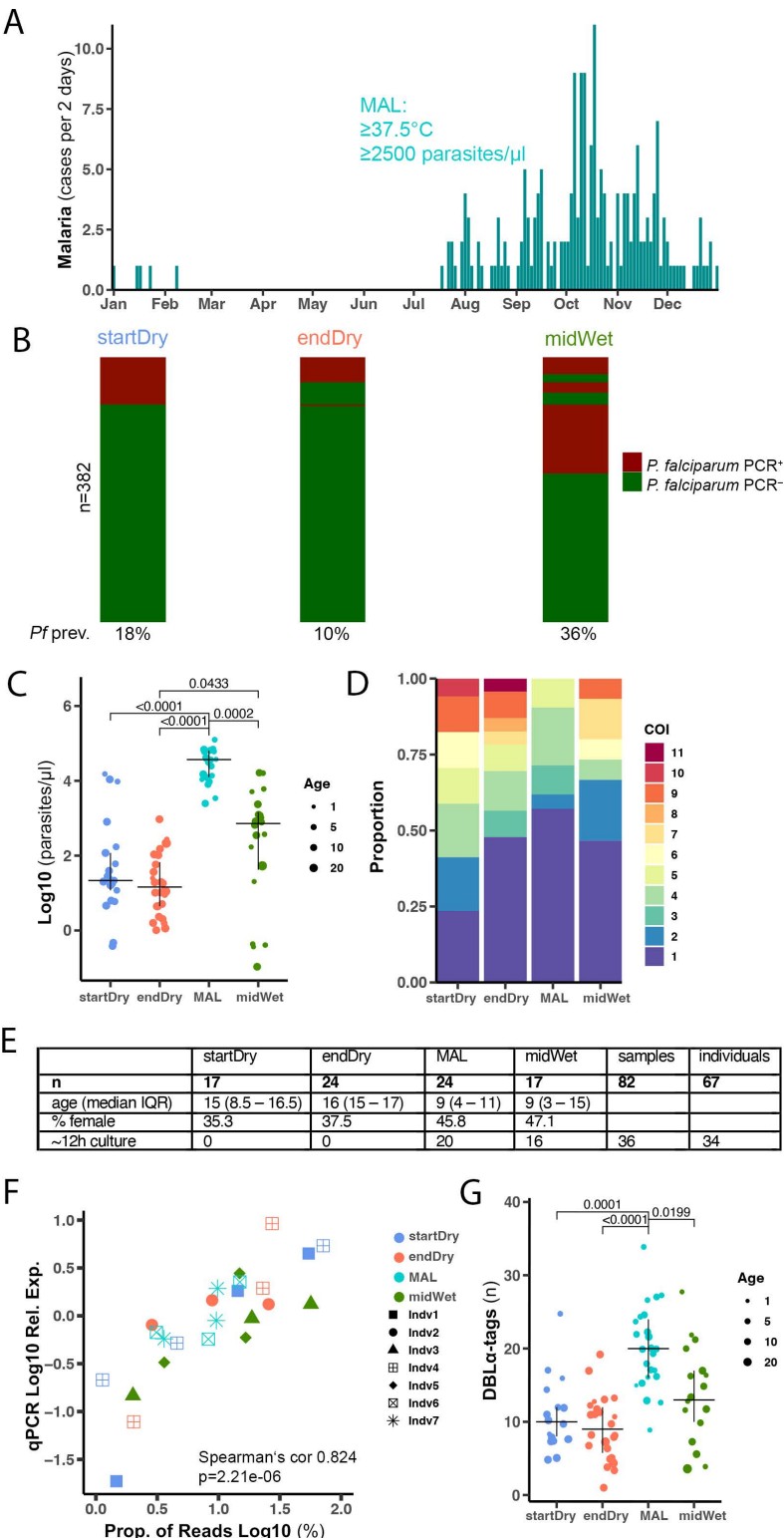

Fig 1. **Expressed DBL α-tag sequencing in a longitudinal cohort in Kalifabougou, Mali. (A)** Clinical malaria frequency in a cohort of ~600 individuals aged 3 months to 45 years measured every 2 days in 2019, diagnosed by axillary temperature ≥37.5 °C and ≥2,500 asexual parasites per µl of blood. **(B)** *P. falciparum* infection prevalence determined by qPCR at the beginning (startDry) and end (endDry) of the dry season, and midway in the wet

season (midWet). Columns are sorted to show the same individual in a single row at all time points. **(C)** Parasitaemia in 104 RDT⁺ samples from startDry (n = 30), endDry (n = 28), and midWet (n = 24) asymptomatic infections, and wet season clinical cases (MAL, n = 22) by qPCR of *var*ATS relative to a standard curve. **(D)** Proportion of individuals with different complexities of infection (COI) measured as number of *P. falciparum ama1* haplotypes in 79 asymptomatic infections (startDry n = 20, endDry n = 23, midWet n = 15) and 21 clinical malaria cases p-value = 0.008 **(E)** Demographics and culture time of samples passing QC in DBLα-tag sequencing **(F)** DBLα-tag read abundance in samples of 7 individuals compared to abundance of the same transcript sequences measured by qPCR relative to a housekeeping gene (Fructose-bisphosphate aldolase, PF3D7_1444800), colour highlights timepoint of collection and shape indicates individual. **(G)** Number of different DBLα-tags observed (0.96 similarity) in clinical cases (n = 24), and asymptomatic infections (startDry n = 17, endDry n = 24, midWet n = 17). C & G Data shows median and IQR, dot size indicates participant age. C, D and G Kruskall-Wallis test with Bonferroni multiple comparison correction.

those with asymptomatic infections in dry (startDry median 14 years, IQR 3 – 16, p = 0.0049, and endDry median 15 years, IQR 12 – 16, p < 0.0001) or transmission season (midWet median 11 years, IQR 4 – 16, p = 0.0067, Dunn's Kruskall-Wallis test). Parasitaemia of 104 samples from 86 individuals, depending on genomic DNA availability, was determined by qPCR of multicopy locus *var*ATS [53]. This revealed higher parasitaemia in clinical malaria cases (MAL), intermediate levels in individuals with asymptomatic infections during the wet season (midWet), and lower parasite densities in the dry season samples (startDry and endDry) (Fig 1C), as previously described [54]. Amplicon sequencing of the polymorphic parasite gene *ama1* was used to determine the complexity of infection (COI) of each sample. One to 11 different haplotypes were detected per sample. We observed a higher proportion of monoclonal infections and lower COI in clinical malaria cases (MAL) than in asymptomatic infections (startDry, endDry and midWet) (Fig 1D). The individuals' age correlated negatively to parasitaemia (Spearman's cor -0.39, p = 0.000123) (S1A Fig); and positively to COI, when the analysis was done with all samples of all time-points (Spearman's cor. 0.329, p = 0.00375) (S1B Fig), but significance was lost when correlations were investigated individually within each timepoint (S1A and B Fig).

To conduct DBLα-tag sequencing, we obtained parasites throughout the year from 118 rapid diagnostic test-positive (RDT⁺) samples of 96 individuals, aged 1–29 years. Samples were collected in the beginning (startDry n = 31) and end (endDry n = 28) of the dry season from asymptomatically infected individuals; and in the ensuing wet season from participants exhibiting their first symptomatic malaria episode of the season (MAL n = 29), and from asymptomatic individuals during a cross-sectional timepoint in the middle of the wet season (midWet n = 28). Following RNA extraction of all 118 samples, we assessed RNA content by qRT-PCR of two *P. falciparum* housekeeping genes, and selected 93 samples with sufficient expression for DBLα-tag sequencing (startDry n = 20, endDry n = 28, MAL n = 25, midWet n = 20), of which 60 samples were prepared in duplicates. DBLα-tags were amplified using a pair of degenerate primers [14]. Samples with <500 sequencing reads across available replicates were excluded, resulting in 82 samples from 67 individuals (startDry n = 17, endDry n = 24, MAL n = 24, midWet n = 17, Fig 1E) with 510–107496 reads per sample (median 27467, IQR 14729 – 46464).

Sequencing tags from all 82 samples were pooled and clustered by at least 96% sequence similarity, resulting in 917 unique DBLα-tags (S1 Table). Technical variation between replicates was assessed comparing samples with PCR replicates each comprising > 500 reads (n = 60). Replicates showed a very strong correlation in reads mapping to specific DBLα-tags (Pearson's cor 0.97, p < 2.2e-16, S1C Fig). Of the variants found in the 60 duplicated samples, less than 2% of DBLα-tags were not shared between replicates; hence for all subsequent analyses, replicates, whenever available, were pooled.

The distribution of expressed *var* genes was determined by quantifying the number sequence reads mapping to each of the 917 DBLα-tags. After removal of DBLα-tags with <1% of the total read count in a sample, we obtained a range of 1–35 different DBLα-tags per infection (median 13, IQR 8–18). To validate the DBLα-tag sequencing, we designed three primer-pairs targeting highly and lowly expressed DBLα-tags in eight samples and quantified the *var* transcript levels by qRT-PCR. The sequenced read counts correlated with the qRT-PCR quantification (Fig 1F, Spearman's cor 0.824, p = 2.21e-6), confirming that DBLα-tag sequencing could identify *var* gene sequences with good representation of their proportional expression.

 

Then, we investigated whether DBLα-tag sequences were uniformly distributed throughout the year. We found significantly higher number of different DBLα-tags per sample in clinical malaria cases (MAL) than in dry season asymptomatic infections (startDry, endDry); and a higher number of different DBLα-tags per sample in wet season asymptomatic infections (midWet) compared to the asymptomatic infections from the end of the dry season (endDry) (Fig 1G). First, to rule out a possible sequencing-based systematic bias, we assessed the sequencing reads and found no differences in read counts between timepoints (S1D Fig). Additionally, we examined whether higher numbers of DBLα-tags were driven by more sequencing reads, but found only a weak negative association between read count and number of DBLα-tags per sample (Pearson's cor -0.265, p = 0.016, S1E Fig). The number of DBLα-tags per sample correlated negatively to the individuals' age (Pearson cor -0.53, p = 3.025e-07, S1F Fig). However, when timepoints were individually considered, the significant negative correlation between number of DBLα-tags and age was observed only among asymptomatic infections at the start of the dry season (startDry, Spearman's cor 0.57, p = 0.018, S1F Fig).

To assess whether progression through the 48 h IDC led to systematic changes in DBLα-tag expression, we sequenced DBLα-tags on 20 malaria cases (MAL) and 16 asymptomatic infections during the wet season (midWet) following ~12h of in vitro culture to compare with the corresponding ex vivo data of the same samples. We observed a high agreement between ex vivo and ~12h-cultured samples, with a median of 77.5% (IQR 61–84) and 82% (IQR 67–89) of DBLα-tags shared in the asymptomatic and clinical cases, respectively (S2 Fig). DBLα-tags lost during the 12h of culture showed significantly lower expression levels in the corresponding ex vivo sample (median 1.7% of reads IQR 1.3 - 2.7) than those DBLα-tags maintaining detection (median 3.3% of reads IQR 1.8 - 6.5, p < 0.0001 Kruskall-Wallis test). This suggested that differences between ex vivo and 12h-cultured samples likely resulted from selection of sub-populations of parasites during culture, or differences in detection of low abundance DBLα-tags, and likely not due to variations of developmental stage between dry and wet season samples impacting the DBLα-tag profiles.

## Parasites express more *var* genes in malaria cases than in dry season asymptomatic infections

The number of DBLα-tags seen per sample correlated positively with parasitaemia (Spearman's cor 0.46, p = 2.54e-5, Fig 2A), but no association between the number of DBLα-tags and the sample's COI was observed (Fig 2B). When timepoints were considered separately, we observed a significant positive correlation between number of DBLα-tags and parasitaemia among asymptomatic infections from the start of the dry season (startDry, Spearman's cor 0.57, p = 0.017, Fig 2A), while other timepoints did not afford statistically significant associations. Among clinical malaria cases, the number of DBLα-tags was positively correlated with COI (Spearman's cor 0.52, p = 0.017, Fig 2B).

We then investigated proportions of different DBLα-tags identified. This showed that the most abundant DBLα-tag in each clinical malaria case (MAL) comprised a lower proportion of reads compared to the most abundant DBLα-tags in dry season asymptomatic infections (startDry, endDry) (Fig 2C). Accordingly, a higher number of DBLα-tags was required to attain 65% of the DBLα-tag reads in malaria cases compared to asymptomatic infections (S3A Fig). This suggests that the parasite population in clinical malaria cases express many PfEMP1 variants at similar proportions, compared to asymptomatic dry season infections, where a single or few *var* genes dominate expression. Wet season asymptomatic infections showed an intermediate state between the two extremes. Indeed, when we calculated the Simpson Diversity Index of each sample based on the proportions of different DBLα-tags, we found significantly higher values in wet season clinical cases (MAL) compared to dry season asymptomatic infections (startDry, endDry) (S3B Fig).

Our approach allowed tracking of DBLα-tags, as well as *ama1* haplotypes across samples within the population. 131 of 917 DBLα-tags occurred in more than one sample, most of those (n = 102) were found in two samples; while the most frequent DBLα-tag was found in 20 samples and likely corresponded to *var1*, a conserved, truncated *var* pseudogene (S1 Table). We did not find an association between samples sharing a DBLα-tag and a particular *ama1* haplotype (Chi-squared, p = 0.98); and we determined that pairs of samples within a timepoint were not more likely to share a DBLα-tag than pairs of samples from different times of the year (Chi-squared, p = 0.73).

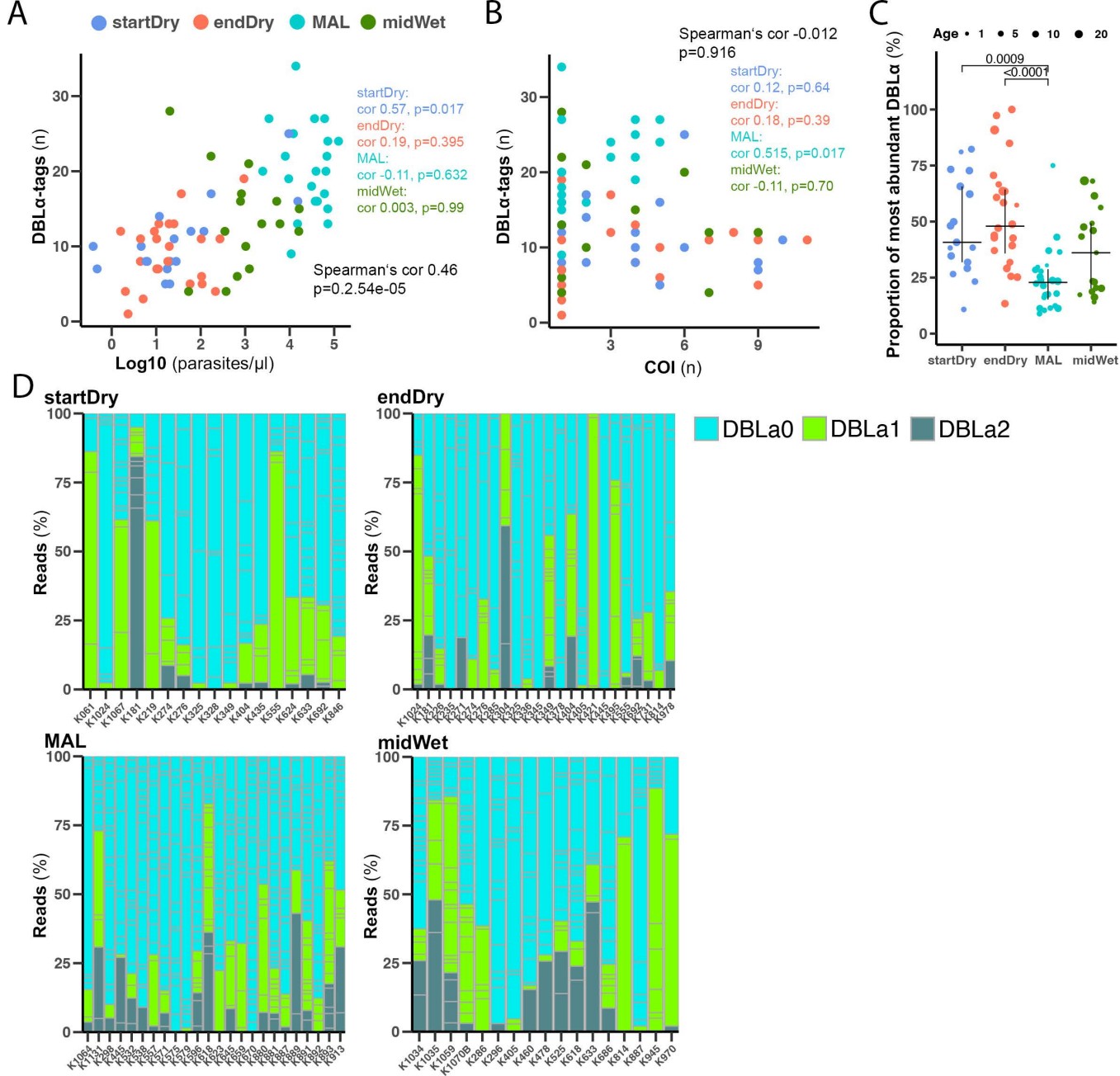

**Fig 2. Malaria cases express more *var* genes than dry season asymptomatic infections.** **(A)** Correlation between number of DBLα-tags and parasitaemia measured by *var*ATS-qPCR. **(B)** Correlation between the number of DBLα tags and COI measured by *ama1*-amplicon sequencing. **(C)** Proportion of most abundant DBLα-tag at the start (startDry n = 17) and end (endDry n = 24) of the dry season, in clinical cases (MAL n = 24), and midway in the wet season (midWet n = 17). Median and IQR are shown, dot size shows participant age, Kruskall-Wallis test with Bonferroni multiple comparison correction. **(D)** Individual distribution of DBLα-type annotated by best Blast hit in a reference *var* database [56]. Each column represents a person, column sub-divisions show individual DBLα-tags.

Next, we used a database of *var* genes [18] to annotate our DBLα-tag sequences to DBLα types 0, 1 and 2 [27] by BLAST comparison, based on their best DBLα hit in the database. We found that DBLα0 was the most represented DBLα type at all timepoints, followed by DBLα1 and DBLα2 types (Figs 2D, and S3C), which is in line with the three type's proportion in *P. falciparum* genome [27]. Comparing the reads counts to the different DBLα types, we observed high variability between samples (Fig 2D), but overall, DBLα0 *var* genes were the most abundant, followed by DBLα1 and DBLα2, and no statistically significant difference was seen among the proportion of the three DBLα types across the timepoints (S3D Fig).

For nine of the individuals analysed, paired samples were collected at the beginning (startDry) and end (endDry) of the dry season, when re-infection is unlikely [8,48], prompting us to investigate *var*-gene expression longitudinally. First, we interrogated COI and clonal persistence between the two timepoints using *ama1* sequencing data. Out of eight individuals where *ama1* amplicon sequencing data was obtained at both timepoints, COI was maintained in paired samples over the course of the dry season (startDry 2.5 IQR 2–5; endDry 2 IQR 1–3, Wilcoxon matched-pairs signed-rank test, p = 0.66). Importantly, in six of the eight individuals (75%), shared clones were detected between the two time points, while random chance of haplotype sharing was ~ 26%. In four out of the eight samples, *ama1* haplotypes found at the end of the dry season were all already detected at its beginning (S4A Fig). Still, four sample-pairs we also identified potential appearances of new haplotypes at the end the dry season (S4A Fig), which may be attributed to the limit of detection of the method used due to low parasitaemia, and/or cytoadhesion of particular parasite clones at the time of blood draw [55]. DBLα-tag comparisons between samples of the same individual at the beginning and end of the dry season revealed a comparable number of DBLα-tags at the two dry season extremes (startDry 8 IQR 8–11, endDry 12 IQR 10–13, S4B Fig), without significant changes in paired samples (Wilcoxon matched-pairs signed-rank test, p = 0.23). However, the DBLα-tag expression patterns were highly divergent between paired samples of the two timepoints, indicative of *var* gene switching. In fact, only in one individual (K181, 11.1%), the same DBLα-tag was detected at both time points (S4B Fig), similar to the ~ 13% chance of finding the same DBLα-tag in two randomly picked samples. In the four paired samples where the *ama1* data revealed that all clones seen at the end of the dry season were present in its beginning, we did not observe any shared DBLα-tags, and no significant difference in the number of DBLα-tags between the two timepoints was observed (S4B Fig, startDry range 8–14, endDry range 10 – 13, Wilcoxon matched-pairs signed-rank test, p = 0.41), supporting switching of the *var* genes expressed.

## Similar *var* gene types and PfEMP1 binding phenotypes predicted in asymptomatic infections and clinical malaria cases

Next, we predicted the domain composition C-terminal to the DBLα domain using Varia [56], and the *var* UPS group using cUPS [57] and the new tool upsAI (Thomas Otto personal communication and https://github.com/sii-scRNA-Seq/upsAI). These bioinformatic tools use DBLα-tag sequences and databases of annotated *var* genes to make probabilistic or machine learning predictions about the *var*-encoded domains and UPS region. First, *var* genes were analysed by their mutually exclusive binding phenotypes defined by their predicted N-terminal CIDR domain. *var* transcripts predicted to encode a CIDRα1 domain were classified as EPCR-binding [32,33], and as CD36-binding if a CIDRα2–6 domain was predicted [58]. *var* genes predicted to encode CIDRβ, γ or δ domains and var1 were classified as having an unknown binding phenotype (n = 107, Fig 3A). For 20% (n = 229) of the DBLα-tags the CIDR domain could not be predicted. This analysis revealed no difference in expression of binding phenotypes between any of the sampling times (Fig 3B). Despite heterogeneity between samples, predicted CD36 binding *var* genes were the most abundant at all timepoints, followed by EPCR-binders (Fig 3A). We then used Varia to predict domains downstream of the first CIDR to identify putative ICAM-1-binding PfEMP1s. These were predicted by the presence of CIDRα1 and DBLβ1/3, and CIDRα2–6 and DBLβ5, representing group A and group B ICAM-1-binders, respectively [59]. Putative ICAM-1-binders were very low or absent at all time points, but we detected statistically significant higher expression of group-B ICAM-1-binders in clinical malaria compared to end-dry season infections (S5A Fig). We also quantified expression levels of *var*s encoding DBLδ and DBLγ

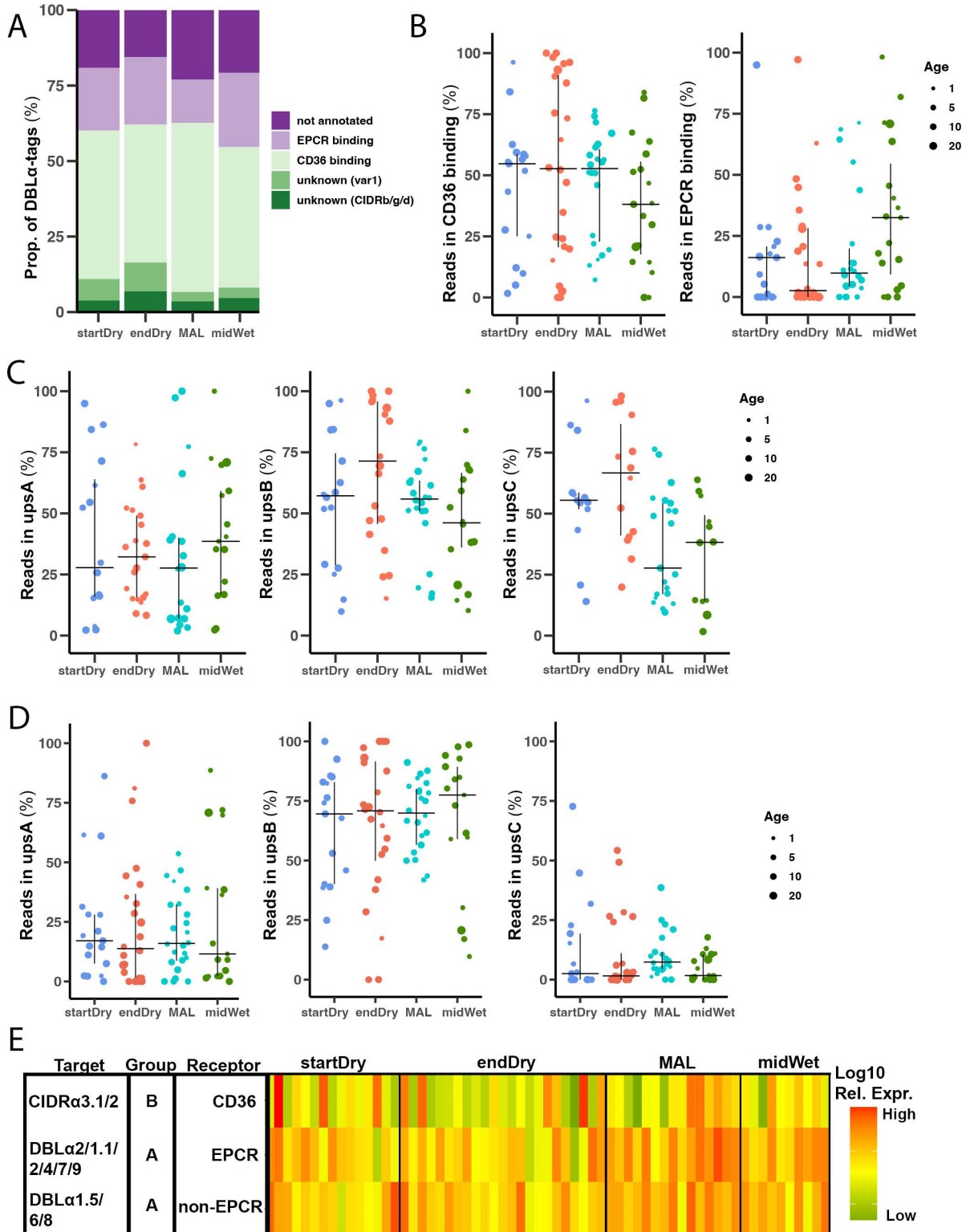

**Fig 3. Similar *var* gene types and PfEMP1 binding phenotypes predicted in asymptomatic infections and clinical malaria cases. (A)** Proportion of DBLα-tags annotated to PfEMP1 binding type based on domain composition predicted by Varia [56]. Genes were classified as EPCR-binding if the N-terminal CIDR domains was predicted as a non-*var1* CIDRα1 domain, as CD36-binding if a CIDRα2 – CIDRα6 domain was predicted, and as unknown, if prediction corresponded to *var1* or a CIDRβ, γ or δ was predicted. **(B)** Proportion of DBLα-tag reads mapping to *var* genes predicted by

Varia to encode CD36- (left) or EPCR- (right) binding PfEMP1s in the beginning (startDry n = 17) and end (endDry n = 24) of the dry season, during clinical malaria (MAL n = 24), and midway (midWet n = 17) in the wet season. **(C)** Proportion of DBLα-tags mapping to predicted group A (left), B (centre), and C (right) *var* genes using cUPS in the beginning (startDry n = 17) and end (endDry n = 24) of the dry season, in clinical malaria (MAL n = 24), and midway (midWet n = 17) in the wet season using cUSP. **(D)** Proportion of DBLα-tags mapping to predicted group A (left), B (centre), and C (right) *var* genes using cUPS in the beginning (startDry n = 17) and end (endDry n = 24) of the dry season, in clinical malaria (MAL n = 24), and midway (midWet n = 17) in the wet season using upsAI. **(E)** Normalized expression of *var* subtypes in the beginning (startDry n = 15) and end (endDry n = 24) of the dry season, malaria cases (MAL n = 15), and wet season asymptomatic infections (midWet n = 10) by qRT-PCR. DBLα2/1.1/2/4/7/9 MAL vs endDry (p-value 0.0083. Median and IQR are shown, dot size show participant age; Kruskall-Wallis with Bonferroni correction.

domains (S5B Fig). As expected, *var*s encoding DBLδ domains found in most *var*s were highly expressed at all timepoints, whereas *var*s predicted to encode DBLγ domains were rare, though a large proportion (45%, n = 504) of DBLα-tags remained unannotated.

We used the cUPS and upsAI tools to assign 68% (n = 772) and 99% (n = 1112), respectively, of the DBLα-tags and to UPSA, B or C (S5C Fig). Both tools predicted similar tags as UPSA *var* genes and gave similar proportions for UPS subtypes but disagreed about assignment of genes between UPSB and UPSC groups. Neither tool resulted in significant differences in abundance between timepoints. In line with genomic proportions [18], UPSB genes were the most expressed at all timepoints, followed by UPSA and UPSC (Fig 3C and D).

To complement the prediction of the domain type composition, we measured expression of a subset of *var* domain types using *var* type-specific primers directly by qRT-PCR in 66 samples (startDry n = 16, endDry n = 24, MAL n = 16, midWet n = 10) for which we also sequenced DBLα-tags. We quantified levels of loci encoding CIDRα3.1/2 domains as a proxy for CD36-binding *var*s, DBLα2/1.1/2/4/7/9 for group A EPCR-binding *var*s, and DBLα1.5/6/8 for group A non-EPCR binding *var*-genes. We observed large variability between samples and mostly non-statistically significant differences in expression level of individual *var* gene types between samples collected in the dry season (startDry and endDry) or during clinical malaria cases in the wet season (MAL) (Figs 3Eand S5D), in agreement with the DBLα-tag observations. The only significant difference observed was a higher expression level of group A EPCR-binding *var*s in clinical malaria cases compared to infections at the end of the dry season (S5D Fig). Comparing qRT-PCR measurements and Varia domain predictions in the same samples, we found a significant correlation between relative expression by qRT-PCR and the percentage of reads matching same domain type in DBLα-tag sequencing (CD36-binder: Spearman's cor 0.492, p = 0.000182; EPCR-binder: Spearman's cor 0.323, p = 0.0183; non-EPCR-binder: Spearman's cor 0.441, p = 0.00096, S5E Fig). We also quantified *var2csa* expression by qRT-PCR and detected increased transcript levels in samples of clinical malaria cases in the wet season (MAL n = 15) compared to asymptomatic timepoints of the dry (startDry n = 4, endDry n = 11) and the wet season (midWet n = 10). However, the high *var2csa* level in ex vivo malaria cases was not observed after 12h of in vitro culture of the sample (MAL 12h) (S5F Fig), suggesting that it may be driven by differences in parasite staging [60].

## Asymptomatic parasites have lower total *var* transcript levels than clinical malaria parasites

We then investigated the overall *var* transcript level per parasite in clinical malaria cases compared to asymptomatic infections using qRT-PCR primers targeting the acidic terminal segment (*var*ATS) [61] in 66 samples collected along the year (startDry n = 17, endDry n = 16, MAL n = 19, midWet n = 14). This showed a significantly higher level of *var*ATS transcripts in clinical malaria cases (MAL) compared to asymptomatic infections from the end of the dry season (endDry) and mid of the wet season (midWet), and also compared to all asymptomatic samples combined (Fig 4A). Additionally, we quantified expression of *ruf6*, a non-coding RNA known to regulate transcription of the *var* gene family [62], and observed similar trends to those seen for *var*ATS quantification, albeit without statistical significance between any of the individual groups; and with borderline significant differences between *ruf6* in clinical malaria cases (MAL) versus all asymptomatic samples combined (Fig 4B). A strong correlation between *ruf6* expression and the expression of *var*ATS was detected (Spearman's cor 0.37, p = 0.0073, Fig 4C). Expression of *var*ATS and *ruf6* did not significantly correlate with the number of DBLα-tags

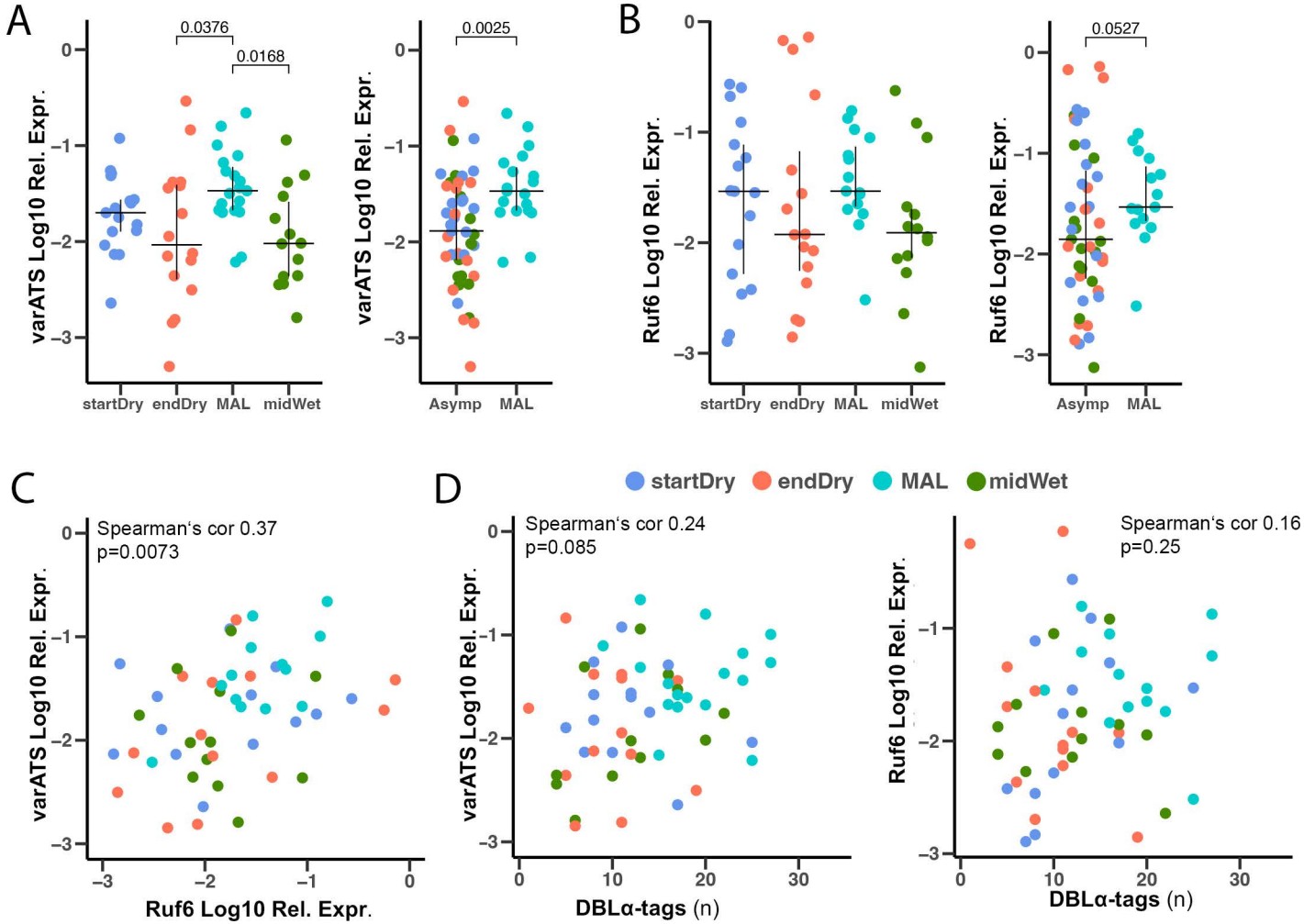

**Fig 4. Asymptomatic parasites show lower *var* transcript levels than clinical cases.** (**A**) *var*ATS relative expression in asymptomatic infections (startDry n = 17, endDry n = 16, midWet n = 14) and malaria cases (MAL n = 19), measured by qRT-PCR. Samples are divided by time point (left) and clinical status (right). (**B**) Relative expression of ncRNA *ruf6* in asymptomatic infections (startDry n = 17, endDry n = 16, midWet n = 14) and malaria cases (MAL n = 19) measured by qRT-PCR. Samples are divided by time point (left) and clinical status (right). Median and IQR are shown, Kruskall-Wallis test with Bonferroni correction. (**C**) Correlation of *var*ATS and *ruf6* expression. (**D**) Correlation of number of DBLα-tags expressed and relative expression of *var*ATS (left) and *ruf6* (right).

found (Spearman's cor *var*ATS 0.24, p = 0.085, *ruf6* 0.16, p = 0.25, Fig 4D), suggesting that the difference in number of DBLα-tags between samples collected at different times of the year was not explained by differences in total *var* gene expression per iRBC.

## Antibody recognition of iRBCs correlates negatively with the number of expressed DBLα-tags

We hypothesized that the more restricted diversity of expressed *var* genes in asymptomatic carriers, compared to clinical cases, was linked to broader and more developed acquired immunity in the older study participants with asymptomatic infections. While we could not quantify antibody responses to the specific PfEMP1 encoded in individual infections, we used three methods to assess humoral immunity to *P. falciparum* antigens. We measured antibodies to AMA1 (a known

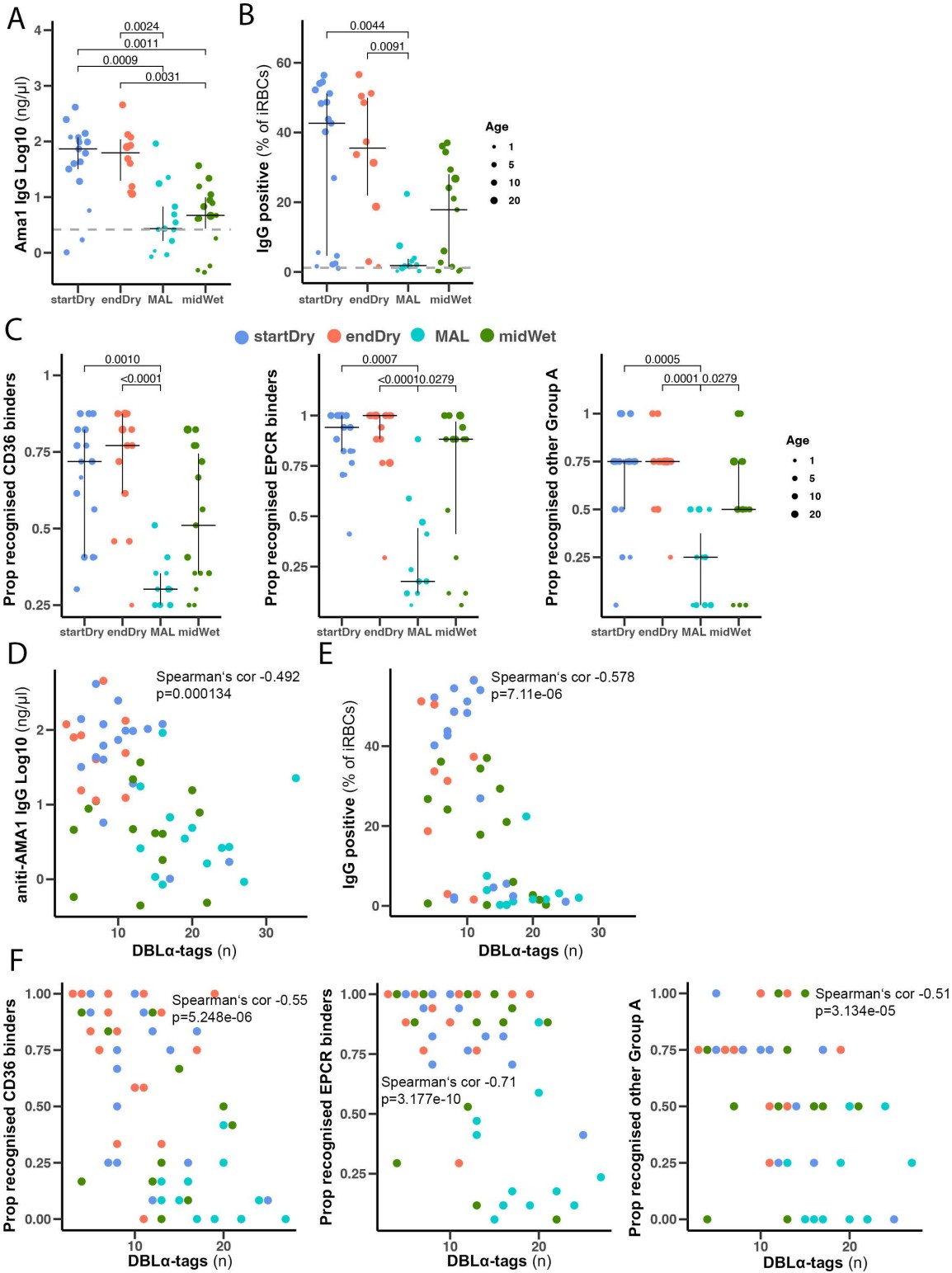

**Fig 5. Antibody recognition of iRBC VSAs correlates negatively with the number of expressed DBL α-tags. (A)** Plasma anti-AMA1 antibody levels in 54 samples (startDry n = 17, endDry n = 10, midWet n = 15, MAL n = 12) measured by ELISA. **(B)** Surface recognition of PfFCR3 iRBCs by plasma of 54 samples (startDry n = 17, endDry n = 10, midWet n = 15, MAL n = 12) measured by flow cytometry. Gray dashed line represents average signal from

naïve plasmas (n = 2) **(C)** Proportion of individuals with antibodies specific to PfEMP1 domains from asymptomatic individuals collected at start (n = 17), end of the dry season (n = 16), clinical malaria cases (n = 11) and mid wet season asymptomatic infections (n = 15). Recognition defined as MFI greater than the level in 6 malaria-naïve controls + 2sd in Luminex. Kruskall-Wallis test with Bonferroni multiple comparison correction. **(D, E)** Correlation of plasma antibody levels measured by AMA1 ELISA **(D)** and Surface recognition assay **(E)** with number of DBLα-tag in the same sample. **(F)** Correlation of proportion of domains recognised and number of DBLα-tags in the same samples.

marker of cumulative malaria exposure [63]) by ELISA, antibodies to VSAs on PfFCR3 iRBCs predominantly expressing the CD36 and ICAM1-binding IT4VAR16 PfEMP1 variant through a flow cytometry-based surface recognition assay (SRA, gating strategy in S6A Fig), and antibodies to 35 PfEMP1 CIDR domains representing CD36 (n = 12) and EPCR (n = 17) binding CIDR domains and n = 6 other CIDR of group A PfEMP1, through a multiplex bead array [64]. Plasma from asymptomatic dry and wet season samples (startDry n = 17, endDry n = 10, midWet n = 17), and from clinical malaria cases in the wet season (MAL n = 10) revealed anti-AMA1 antibodies in most individuals, with significantly higher titers in those maintaining asymptomatic infections during the dry season compared to individuals with clinical malaria or asymptomatic infections in the wet season (Fig 5A). Consistent with the hypothesis that dry season asymptomatic carriers have an increased ability to restrict PfEMP1s, we detected higher surface recognition of VSAs (Fig 5B), and a broader recognition of CD36-, EPCR- and other group A PfEMP1-binders (Fig 5C) in plasma of the dry season asymptomatic infections than in those of clinical malaria cases. Titres of anti-AMA1 antibodies, levels of iRBC surface recognition, and the breadth of the anti-PfEMP1 immune response were highly correlated to each other (S6B, C Fig) and associated with age (S6D Fig). Plasma AMA1 antibody titers, iRBC surface recognition, and broader recognition of PfEMP1 domains of any binding phenotype correlated negatively with the number of DBLα-tags detected in its sample (AMA1: Spearman's cor -0.492, p = 0.0001; SRA: Spearman's cor -0.592, p = 2.41e-6, CD36 binders: Spearman's cor -0.55 p = 5.248e-06, EPCR binders: Spearman's cor - 0.71p = 3.177e-10, other group A Spearman's cor -0.51 p = 3.134e-05) (Fig 5D-F).

## Single-cell RNAseq data supports restricted expression of *var* genes in dry season asymptomatic infections

Finally, we used single-cell SMART-Seq to characterize *var* transcripts and domain architecture obtained from RNA libraries of single FACS-sorted iRBCs of one clinical malaria case (MAL) and one end of the dry season asymptomatic infection (endDry) collected in 2020. Contrasting to DBLα-tag sequencing, SMART-Seq generates reads across the whole *var* gene [65], and hence allowed detecting domains downstream of DBLα without bioinformatic predictions. Both samples were processed ex vivo, and parasites from the clinical malaria case was additionally examined following ~16h of in vitro culture (MAL16h). A total of 115 single FACS-sorted iRBCs (endDry n = 54, MAL n = 31, MAL16h n = 31) were sequenced, and the reads obtained were mapped to a combined *P. falciparum* and human reference. After quality filtering, 111 iRBCs with >100 *P. falciparum* genes and >1000 *P. falciparum* reads, remained for further analyses (endDry n = 51, MAL n = 29, MAL 16h n = 31). First, we assessed whole transcript nucleotide variation across all variant and invariant sites in *P. falciparum* genome using GATK4 [66] to determine the pairwise nucleotide similarity between iRBCs. By hierarchical clustering, iRBCs of the clinical malaria case, independent of culturing, clustered with each other and apart from iRBCs from the end of dry season infection, indicative of one clone in the clinical malaria case and another in the end of the dry season asymptomatic infection (Fig 6A). Parasites were staged by comparison to a single-cell atlas [67], reproducing previously described circulation of older stages in dry-season asymptomatic infections [8]. We then mapped the 111 quality-filtered iRBC transcriptomes to the annotated *var* database of 200.000 *var*s and one million *var* domains [18] used by the Varia and cUPS tools, as limited sequencing coverage precluded *de-novo* assembly of the full *var* genes. Cells were defined as *var*-expressing when more than three reads mapped to the *var* reference and coverage of domains other than the ATS were detected, resulting in 48 and 41 *var*-expressing iRBCs in the dry season and in clinical malaria samples, respectively. We further clustered cells based on *var* domain expression and found 7 clusters of iRBCs expressing similar *var* domains in the end of the dry season iRBCs (Fig 6B) and 22 in the clinical malaria case (Fig 6C), which likely correspond

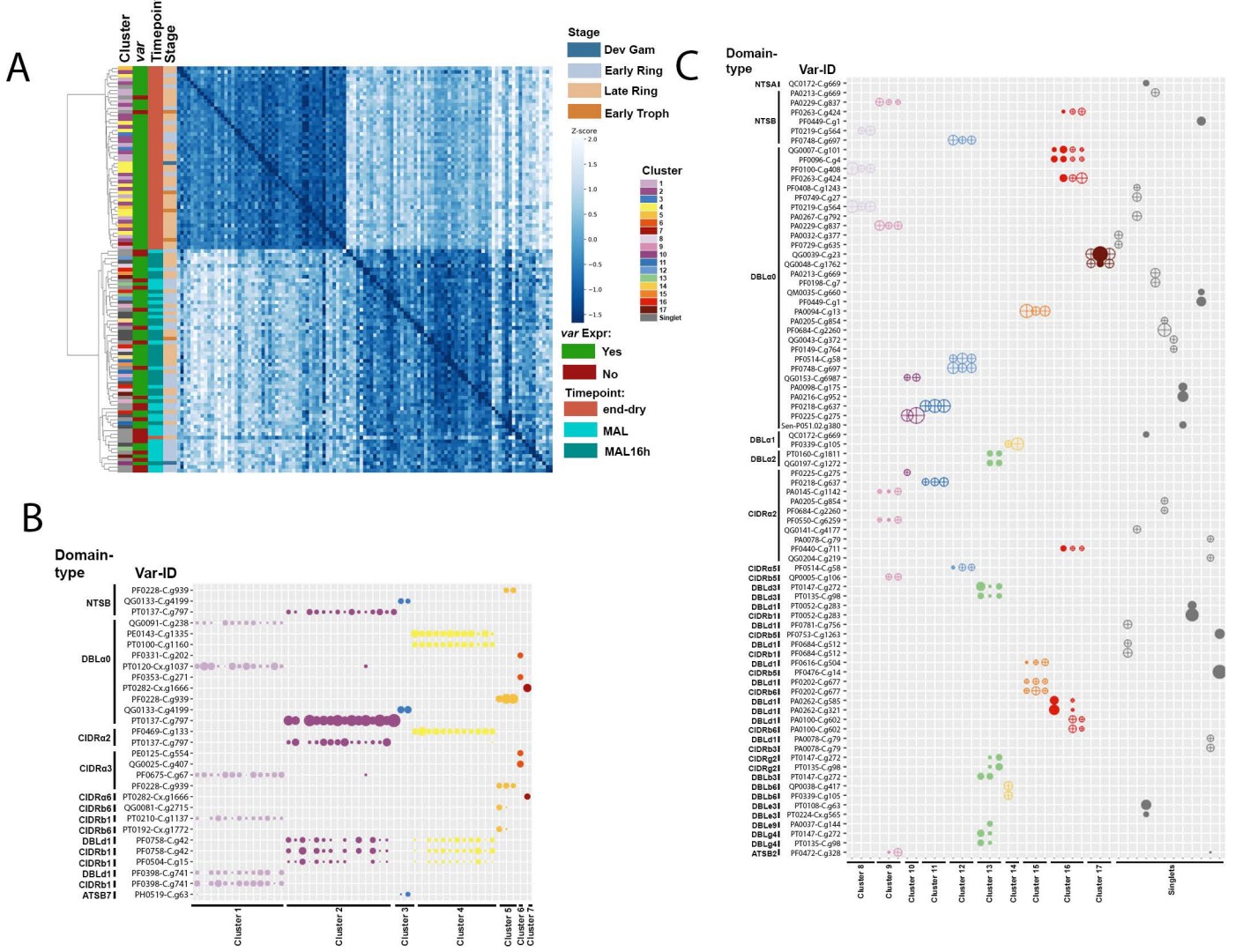

**Fig 6. Single Cell RNAseq data supports restricted expression of *var* genes in dry season asymptomatic infections. (A)** Heatmap and hierarchical clustering of z-score values of whole transcriptome nucleotide similarities between single iRBCs of one dry season sample (end-dry, n = 51) and one malaria case (MAL, n = 29), that additionally was in vitro cultured for ~16h (MAL16h, n = 31). Bars next to the heatmap indicate predicted stage, collection timepoint, var expression status and var cluster for each cell. **(B)** Summary of *var* domain coverage in *var*-expressing cells of *var* domains with >1000 normalized reads in at least one cell within the dry season sample (n = 48). **(C)** Summary of *var* domain coverage in *var*-expressing cells of *var* domains with >1000 normalized reads in at least one cell of the malaria case' sample (n = 41). Cells with similar *var* domain coverage are clustered, cells sequenced ex vivo and after 16h of culture are indicated by circles and crosses, respectively.

to groups of cells, each expressing a shared *var* gene. This highlighted a higher number of *var* domain clusters in the clinical malaria case (5.4 per 10 iRBCs), compared to a lower number of *var* domain clusters per 10 iRBCs in the end of the dry season sample (1.5 per 10 iRBCs). Similarly, 50% of iRBCs covered a minimum of 2 different *var* domain clusters in the dry season sample, while they covered at least 7 different *var* domains in the clinical malaria case, indicating a higher diversity of *var* gene expression in clinical malaria compared to end of the dry season, as was found by DBLα-tag sequencing (Figs 1 and 2).

For most *var* domain clusters, we detected DBLα, CIDRα and further downstream domains, though in most clusters at least some domains of a full *var* gene were missing (S7A-C Fig). In several clusters, we observed mapping to multiple *var*

domains of the same domain type, which could appear inconsistent with monoallelic expression. We observed 11 of these instances in clusters of the end-dry season sample, and 24 instances in the clinical malaria case's clusters. In *var* cluster 1 of the end-dry season sample, we identified reads of one iRBC mapping to two distinct DBLα0 sequences, however, the coverage of these two domains did not overlap, suggesting that the sequenced DBLα domain was a combination of separate DBLα sequences found in the database (S7A Fig). More frequently, however, there was ample overlap in the covered regions of similar domain types, indicating multiple highly similar *var* domains in the database, indicating that the specific *var* genes circulating in the samples are not present in the database used for *var* gene mapping, hence our reads mapped to multiple *var* domains of the same domain type within highly similar reference sequences. Indeed, in seven out of ten instances in the end of the dry season clusters (S7A and B Fig), and in all instances in the clinical malaria case' clusters (S7C Fig), pairwise sequence comparison by BLAST revealed high sequence similarity of the covered regions within each instance (median: 99.85%, IQR 98.67-100.00 for the end of the dry season, and median: 99.61%, IQR 96.32-100.00 for the clinical malaria case). In three further instances of mapping to different domains of the same domain type in the end of dry season sample, coverage of one domain type variant was found in the majority of cells in the respective cluster, including a single cell which exclusively also showed lower coverage of another domain type variant. This other domain type variant was also found covered in multiple cells of another cluster with high coverage, suggesting minor cross-contamination between few cells (S7A Fig). All DBLα domains identified in the dry season sample were DBLα0, while in the malaria case all three types of DBLα were seen, with a minority of DBLα1 and 2, and the majority of DBLα0s (Tables S2 and S3). We found CIDRα2.3/5/8, 3.1/2 and 6 at the end of the dry season, while in the clinical malaria case CIDRα2.1/2/3/4/9, 3.4, 4, and 5 domains was found (Tables S2 and S3).

## Discussion

Studies of *var* expression in clinical cases have linked genes encoding EPCR-binding PfEMP1 to severe malaria, and those encoding CD36-binding PfEMP1 to uncomplicated clinical malaria [40,41,68–70]. However, if and how specific *var* gene and PfEMP1 subsets play a role in the persistence of infection in asymptomatic hosts remains poorly understood. Ruybal-Pesántez and colleagues sequenced DBLα domains of over 1000 parasite genomes of asymptomatic individuals in Ghana, identifying more than 40000 DBLα sequences without reaching coverage saturation of the full *var* gene population, and reported high DBLα diversity in both the wet and the dry seasons [71], suggesting that seasonal selection at the genomic level was unlikely, but expression patterns were not investigated. In this study, we examined expression of *P. falciparum var* genes through sequencing of expressed DBLα-tags in 82 samples of 67 Malian children. We identified 917 different DBLα-tags (*var* genes) and validated the relative and absolute transcript levels using primers specific to individual DBLα-tags and the shared *var* exon2 region (varATS qRT-PCR). Overall, parasites from asymptomatic individuals expressed fewer DBLα-tags (Fig 1), showed higher dominance of the most expressed *var* gene (Fig 2) - without preference for a specific *var* UPS type or PfEMP1 binding phenotype (Fig 3) - and had lower total *var* transcripts compared to clinical malaria cases (Fig 4).

Detection of a higher number of different *var* transcripts in clinical malaria cases and its correlation with parasitaemia, was unlikely due to PCR bias [72], as this was minimized by ensuring consistent housekeeping gene expression and gel electrophoresis validation, leading to similar read depths (S1D Fig), strong congruency between replicates (S1C Fig), with findings supported by both qRT-PCR (Figs 1F, and S5E) and SMART-Seq single-cell sequencing (Fig 6).

Few studies have analysed the expressed *var* diversity in asymptomatic individuals. Mugasa and colleagues examined full-length *var* mRNA isolated by magnetic beads tagged with a *var*ATS probe, followed by cloning into vectors and sequencing to compare expression between severe malaria and asymptomatic infections of infants [73]. The authors found no difference in the number of distinct DBLα sequences per isolate between the clinical groups. With the same approach, Falk and colleagues investigated *var* expression in severe, uncomplicated, and asymptomatic children from Papua New Guinea. Also they reported a similar number of DBLα sequences between the three conditions [40]. This

method of amplification, cloning, transfection and sequencing may introduce bias, as the number of unique sequences to be discovered depends on the number of clones sequenced, which is limited and variable between samples, especially since saturating coverage within individuals was not met. The method used in the present study more robustly avoids sampling bias, as the number of potential novel sequences is only limited by sequencing depth and presented no correlation between number of DBLα-tags and sequencing read counts.

In addition to the studies by Mugasa et al. [73] and Falk et al. [40], which reported mixed *var* gene types with a predominance of *var*s encoding CD36-binding group B and C genes, only a few studies have attempted to define the *var* and PfEMP1 types expressed in asymptomatic carriers. Kaestli et al. [13] investigated *var* expression in asymptomatic and clinical malaria in Papua New Guinea using qRT-PCR assay targeting *var* UPS and reported increased UPSC levels in asymptomatic compared to clinical cases. The capture method used to enrich *var* transcripts precluded direct normalization of their expression data, and they instead used genomic DNA of cultured lab strains to calculate *var* copy numbers, which may have introduced bias. In Mkumbaye et al., a lower level of transcripts encoding EPCR-binding PfEMP1 was found in asymptomatic Tanzanian children by qRT-PCR targeting these genes [74]. These studies could not quantify the number of expressed *var*s or the total *var* expression level. The combined DBLα-tag sequencing and *var* type prediction approach used in the present study allowed a higher resolution and more accurate annotation of the expressed *var* genes. However, our approach has limitations as it relies on semi-quantitative PCR amplification using degenerate primers and annotated *var* genes [18] from genomes of the Pf3K and MalariaGEN projects [75]. Nevertheless, we could annotate the majority of amplified tags and found no significant difference in *var* UPS type expression between clinical and asymptomatic infections. Although UPSC *var* genes were the most frequently observed in end of the dry season asymptomatic infections and lowest in clinical malaria by cUPS, these were not statistically significant and not corroborated by upsAI (Fig 3D). Similarly, we observed transcripts representing all PfEMP1 phenotypes, albeit with a dominance of genes encoding CD36-binding PfEMP1 in both asymptomatic and clinical cases. These profiles resemble those previously observed in uncomplicated malaria, and indicate that persistent, asymptomatic infections are not linked to separate *var* gene subset or PfEMP1 phenotype, but is likely associated with the immune status of the individual. Indeed, the present asymptomatic carriers were significantly older and had higher levels of antibodies to both merozoite and VSA antigens, including PfEMP1, than individuals with clinical infections (Fig 5). Warimwe and colleagues have shown in Kenyan children more homogeneous *var* expression in asymptomatic infection associated with a broader host antibody response of slightly older children [42].

IgG targeting and inhibiting PfEMP1 contributes to malaria immunity [76]; and controlled [58] and natural infections [77] of less immune individuals have been linked to expression of a higher number of *var* genes, which is restricted in individuals with higher immunity. Protection from severe malaria is rapidly acquired after a few episodes in areas of high transmission [45], likely due to development of cross-reactive antibodies recognizing PfEMP1-variants associated with severe disease [78]. Thus, acquisition of PfEMP1 antibodies recognising PfEMP1 is ordered, with earlier responses to pathogenic EPCR binding PfEMP1 variants [79,80]. Longer periods of multiple infections seem required to accumulate antibodies recognizing PfEMP1 associated with uncomplicated malaria, finally resulting in mostly avirulent infections in clinically-immune adolescents and adults [51,81,82]. The *var* variants expressed in asymptomatic individuals may represent gaps in the immune coverage of those hosts. However, since we observed all types of *var* genes—including those frequently recognized by exposed individuals—expressed in both clinical and asymptomatic infections, it is plausible that parasites persisting asymptomatically over extended periods survive not only by randomly switching to antigenically distinct variants [83], but also by reducing the overall PfEMP1 presentation and cytoadhesion efficiency, rather than relying on specific PfEMP1 types. In support of this notion, we found indications of reduced overall *var* expression in asymptomatic vs clinical malaria cases reported by *var*ATS qPCR primers targeting the semi-conserved and shared *var* exon 2 (Fig 4). Decreased overall *var* expression is also supported by lower the *ruf6* transcripts detected in asymptomatic infections compared to clinical malaria cases (Fig 4). This is in line with findings by Guillochon and colleagues, who assembled de novo *var* gene

expression from RNA-seq of cerebral versus uncomplicated malaria cases and reported lower overall *var* gene expression in uncomplicated cases [84]. Also, a trend of reduced expression of the top *var* gene in asymptomatic end of dry season infections compared to clinical malaria cases described in [8]. Nevertheless, the *var*ATS primers coverage across the *var* family is only ~40%, leaving the possibility of a skewed and not reduced *var* expression. The single cell approach used to validate expression of *var* genes of dry and wet season infections (Fig 6) was also not ideal to corroborate the overall expression level of *var* due to the very small number of cells available and low coverage of *var* genes.

Reduction of adhesin expression and endothelial cell binding capacity in *P. falciparum* has been documented in a splenectomized patient [85], similarly *var* expression can be lost in vitro [86] just as surface expression of PfEMP1 was recently shown to vary independent of *var* transcript level [87].

We also observed higher levels of *var2csa* transcripts in clinical malaria cases compared to dry season asymptomatic infections (S5F Fig). This *var2csa* expression appeared to be transient as it was lost after a short time of in vitro culture. The difference in *var2csa* expression may be driven by differences in parasite stage between clinical and asymptomatic infections, as *var2csa* has also been shown in vitro to peak earlier in its expression than other *var* genes [60]. In vitro, *var2csa* expression depends on the intracellular levels of S-adenosylmethionine (SAM) [88], and its expression has been suggested to play a central role in *var* gene switching [89]; requiring further investigations in natural infections. If, and to what extent, parasites released from the liver include both high and low PfEMP1 expressors, or if such phenotypic variation can arise in blood-stage progeny, remains unclear. Additionally, it is uncertain how low cytoadhesive parasites would survive in individuals with a functioning spleen.

In conclusion, we found that *P. falciparum* parasites from asymptomatic individuals exhibited *var* expression profiles with fewer, more dominant variants compared to clinical infections, likely influenced by host immunity. However, the lower overall *var* transcript abundance and presence of all *var* gene types in asymptomatic carriers suggest that these parasites may represent a subset with particularly efficient switching or reduced cytoadhesion due to lower PfEMP1 presentation.

## Materials and methods

### Ethics statement

The study was approved by the Ethics Committee of Heidelberg University Hospital; the Faculty of Medicine, Pharmacy and Odontostomatology at the University of Bamako; and the National Institute of Allergy and Infectious Diseases of the National Institutes of Health Institutional Review Board, and is registered at ClinicalTrials.gov (identifier NCT01322581). All study participants or their parents/guardians gave written informed consent to acquisition of samples and clinical data.

### Study site and sample collection

Samples and clinical data were obtained in 2019 from a longitudinal cohort study of ~600 individuals aged 3 months to 45 years in Kalifabougou, Mali, as described elsewhere [48,51]. Clinical malaria cases were defined as *Plasmodium* infection with parasitaemia >2500 parasites/µl blood and an axillary temperature >37.5°C and no other apparent cause of fever and were treated according to national guidelines with a 3-day course of artemether and lumefantrine. Study exclusion criteria were haemoglobin <7 g/dl, axillary temperature ≥37.5 °C at enrolment and acute systemic illness or use of antimalarial or immunosuppressive medications in the 30 days preceding enrolment.

Samples used in this study were collected at scheduled cross-sectional timepoints in January, May and October 2019, and in July 2020 as well as during passive surveillance of clinical malaria episodes in 2019 and 2020. At cross-sectional visits and from study participants presenting their first clinical malaria episode during the transmission season 2019, dried blood spots on filter paper (Protein Saver 903, Whatman), thick blood smears and venous blood (4 ml from study participants 4 years or younger, 8 ml from older study participants) were collected. Venous blood was drawn by venipuncture, collected in sodium citrate-containing cell preparation tubes (Vacutainer CPT Tubes, BD) and separated into plasma and

RBC pellets through centrifugation. Subsequently samples were frozen in liquid nitrogen and transported to Heidelberg/Berlin for further analysis.

## Detection of clinical malaria and subclinical *P. falciparum* infection

At sample collection timepoints during cross-sectional visits, venous blood samples were tested for subclinical *P. falciparum* infection by rapid diagnostic test (RDT; SD BIOLINE Malaria Ag P.f test of histidine-rich protein II) prior to freezing. Clinical malaria cases were detected by microscopy in Giemsa-stained thick blood smears by light microscopy. *P. falciparum* infection was confirmed retrospectively by nested PCR for the *P. falciparum* 18S rRNA as described elsewhere [52].

### *P. falciparum* short-term culture

Part of RBC pellets of RDT+ asymptomatic samples at the October 2019 cross-sectional timepoint and clinical malaria cases during the transmission season 2019 were cultured prior to freezing at 2% haematocrit in RPMI 1640 (Gibco) complete medium (with L-glutamine and HEPES), 7.4% sodium bicarbonate (Gibco), 100 µM hypoxanthine (C.C.Pro) and 25mg/ml gentamycin (Gibco) added with 0.25% Albumax II (Gibco) for 12 h at 37°C in a candle jar as previously described [90]. After 12 h of short-term culture, pellets were collected, frozen in liquid nitrogen and shipped to Heidelberg/Berlin.

## Parasitaemia measurement by varATS qPCR

DNA was extracted from frozen RBC pellets using the QIAGEN DNeasy Blood & tissue kit according to manufacturer specifications. qPCR for the varATS locus was performed as previously described [61], using the TaqMan Multiplex Master Mix (Applied biosystems). Quantification was performed based on a standard curve prepared by serial dilution of *in vitro* cultured ring stage *P. falciparum* parasites of a Malian parasite isolate in RBC. Concentrations below 0.1 parasites/µl, the lower limit of the standard curve's dynamic range, were considered below the limit of detection.

### *ama1* amplicon sequencing

DNA was extracted from frozen RBC pellets using the QIAGEN DNeasy Blood & tissue kit. *ama1* amplicon sequencing was performed as previously described [54]. Clusters present in both PCR replicates of a sample and a total read count >500 were included in the analysis.

## RNA extraction and cDNA synthesis

RNA was extracted using the phenol-chloroform method performed over 2 days. Frozen RBC pellets were quickly thawed at room temperature and 750 µl Trizol LS reagent (ambion) added to up to 250 µl RBCs pellet. The RBC pellet with Trizol mixture was thoroughly mixed by vortexing and the resulting emulsion which was immediately spun at 12,000 g for 15 min at 4 °C. The resulting separated top aqueous layer only (without the inter- and organic phase) was transferred into a tube already containing 500 µl Isopropanol and 3 µl GlycoBlue and stored overnight at -20 °C. On day 2, the RNA pellet was separated by spinning tubes for 60 min at 12,000 g and at 4 °C. This is followed by washing twice in cold ethanol and complete drying of the ethanol from the tube. RNA pellet was resuspended in 20 µl RNase-free water. Genomic DNA was removed with Amplification grade DNase 1 (invitrogen) according to manufacturer's instructions. To check for effective removal of genomic DNA, 1 µl of RNA was ran in a qPCR with primers for two *P. falciparum* genes, P61 (fructose-bisphosphate aldolase, PF3D7_1444800) and P90 (serine-tRNA ligase, PF3D7_0717700) using previously designed primers [30,60]. Reactions were set up with Power SYBR Green Master (Applied biosystems), primers were used at a final primer concentration of 1 µM. Cycling conditions were 10 min 95°C, followed by 30 cycles of 30 sec at 95°C, 40 sec and 50°C, 50 sec at 65°C, and a final elongation of 40 sec at 68°C, followed by a melt curve. qPCRs were run on a

QTower3 (Analytic Jena). cDNA was synthesized with the Superscript IV vilo cDNA Reverse Transcription Kit (invitrogen) using 1 µg of template RNA according to manufacturer specifications.

## DBLα-tag sequencing

First, *P. falciparum* transcript abundance in cDNA samples was quantified by qPCR for P61 and P90 housekeeping genes as described above with 2 µl of cDNA as input volume and 40 cycles. Amplification using P61 primers provided higher yield of nucleic acid (lower Ct value) as compared to P90 for the same sample. Therefore, Ct values of P90 primer reactions were used to exclude samples deemed to have too little material for sequencing. Samples with Ct-values >30 for the P90 housekeeping gene were excluded and for samples with Cts between 21–29 subsequent steps were performed in duplicates. As for the samples with Cts of 12 -14.8, the cDNA was diluted 1:100 and for those with Cts > 12 they were diluted 1:1000 prior to subsequent steps. Pre-amplification of DBLα-tag sequencing was performed with the varF_dg2 and Brlong primers. Per reaction, 1 µl of cDNA was mixed with KAPA HiFi fidelity buffer (Roche, 1x final concentration), dNTPs (Roche, final concentration each 0.3 µM), the abovementioned primers (final concentration each 2 µM), at KAPA HiFi Hotstart polymerase (Roche) and run with the following cycling conditions (Table 1).

 2 µl of PCR product were run on an agarose gel to inspect product size and primer dimer noise. The expected amplicon size was 350 – 500 bp, samples without a visible band at the expected size were excluded. After amplification DBLα-tag sequencing samples were pooled and sequenced on an Illumina MiSeq with 2 × 300 paired-end cycle protocol. Only samples with >500 reads across replicates after quality filtering were included in subsequent analysis (Table 2).

## DBLα-tag sequencing bioinformatics

Primer sequences were removed, and adapter and quality trimming performed using trim-galore (https://github.com/Felix-Krueger/TrimGalore?tab=readme-ov-file, v. 0.6.10). Processing and filtering of DBLα-tag sequencing followed the steps of the DBLaCleaner pipeline (https://github.com/UniMelb-Day-Lab/DBLaCleaner/) [91]. Briefly, paired-end reads were

**Table 1. PCR cycling conditions.**

| Step | Temp (°C) | Duration | Cycles |
|---|---|---|---|
| Hot start | 95 | 2 min | 1 |
| Denaturation | 98 | 20 sec | 5X |
| Annealing * | 54 | 30 sec (0.5 °C/sec) ramp | |
| Elongation | 68 | 1 min 15 sec (1°C/sec) ramp | |
| Denaturation | 98 | 20 sec | 30X |
| Annealing | 54 | 30 sec | |
| Elongation | 68 | 1 min 15 sec (1°C/sec) ramp | |
| Final elongation | 72 | 2 min | 1 |

* For the first 5 cycles cooling from denaturation temperature was performed to 65°C at max ramp (3°C/sec), then cooled to 54 °C with 0.5 °C/sec ramp. All other steps ramp 3 °C/sec.

**Table 2. Starting samples and filtering steps.**

| | startDry | endDry | MAL | midWet | samples | individuals |
|---|---|---|---|---|---|---|
| n | 35 | 29 | 26 | 28 | 118 | 95 |
| P61/P90 Ct<30 | 20 | 28 | 25 | 20 | 93 | 75 |
| Visible gel band | 18 | 25 | 24 | 17 | 84 | 70 |
| >500 reads | 17 | 24 | 24 | 17 | 82 | 67 |

merged using PEAR (v. 0.9.6; -v 10 -n 200) [92]. Merged reads from all samples were concatenated, filtered (--fastq_ filter --fastq_maxee 1.0), and overlapping reads collapsed excluding sequences with < 2 reads support (--derep_prefix --minuniquesize 2), and chimeric reads removed (--uchime_denovo) using vsearch v2.15.2 [93]. Remaining sequences were clustered with a 96% nucleotide identity threshold to obtain representative sequence clusters (--cluster-fast –id 0.96). DBLα-tag clusters are referred to as DBLα-tags throughout the manucript. Non-DBLα-clusters were identified with hmmsearch (HMMER3.3.2, hmmer.org) and excluded, the hmmfile was taken from (https://github.com/UniMelb-Day-Lab/DBLaCleaner/). Filtered reads in individual samples were mapped back to remaining DBLα-clusters using vsearch (--usearch_global –id 0.96) to quantify cluster abundance by sample. DBLα-tag sequence clusters were annotated with Varia [56] in GEM mode with default parameters.

Simpson index for a sample was calculated as $1 - \sum (\text{proportion of reads in cluster})^2$. DBLα-tag sequences were translated into amino acids and annotated with cUPS [57] and upsAI (Thomas Otto personal communication and https://github.com/sii-scRNA-Seq/upsAI) in tag mode using default parameters. 0.95 probability cutoff was used to assign clusters to upsA, upsB or upsC groups based on the cUPS prediction.

## qRT-PCR validation of specific DBLα-tag sequences primer sequences design

For a subset of samples primers specific for individual DBLα-tag-sequences were designed using NCBI primer BLAST [94] by providing the sequence cluster of interest as PCR template. In the primer pair specificity parameters, under database, the option "custom (use your own sequence accession, assembly accession, etc)" was selected and the DBLα-tag sequences of the corresponding sample provided as a fasta file. *Plasmodium* (Laverania) *falciparum* (taxid:5833) was selected under organism name for exclusion. All other parameter values remained as default. The same process was carried out for the highest, one medium, and one low abundance sequence cluster per sample. Even though the default number of designed primers is 10, the first 4 primers were tested in the qRT-PCR validation.

## qRT-PCR validation of specific DBLα-tag sequences

For each sequence cluster, 4 different forward and reverse primers were evaluated. All qRT-PCR were performed in 20 µl reaction volume, primer concentrations of 1 µM. Each cluster was quantified relative to P90 and P61 housekeeping genes. For each DBLα-tag sequence cluster, a cDNA master-mix for the total number of reactions was prepared containing Power SYBR-green Master Mix 2X, the total required volume of cDNA and nuclease-free water to make up the volume to 18 µl. The required cDNA volume used was based on the previous P90 housekeeping gene qRT-PCR performed prior to the DBLα-tag sequencing. Samples which had P90 Cts of <15, 1.125 µl cDNA was added per reaction, those with Cts of 15–19 had 1.25 µl per reaction and finally the rest of the samples had previous Cts of 22–26 for which 2 µl of cDNA was added to each reaction. PCR cycling conditions were 10 min at 95 °C, followed by 40 cycles of 30 sec at 95 °C, 40 sec at 50 °C, and 50 sec at 65 °C, with final elongation of 40 sec at 68 °C followed by a melt curve. All qPCRs were run on a QTower3.

For each DBLα- tag cluster, primer pairs which amplified in reactions with only one melting peak were considered to calculate an average gene Ct relative to their average of Ct P90 and P61 genes.

## *var* domain type qPCR

var-domain type qPCR was performed as described in [14]. Primers for the different domain types were pre-mixed as described in Table 3, to a final total primer concentration of 20 µM for forward and reverse primers. Primers sequences are found in the Key Resources Table. PCR cycling conditions were 15 min at 95 °C, followed by 40 cycles of 30 sec at 95 °C, 40 sec at 50 °C, and 50 sec at 65 °C, with final elongation of 40 sec at 68 °C.

**Table 3. Primer mix details for different *var* domain types.**

| *var*- domain type | Primer | FW/ REV | H20 (µl) | primer from 100 µM stock (µl) |
|---|---|---|---|---|
| DBLα1.5/6/8 | DBLα1.5/6/8_fwd 1 | forward | 60 | 10 |
| | DBLα1.5/6/8_fwd 2 | forward | | 10 |
| | DBLα1.5/6/8_rev 1 | reverse | | 4 |
| | DBLα1.5/6/8_rev 2 | reverse | | 4 |
| | DBLα1.5/6/8_rev 3 | reverse | | 4 |
| | DBLα1.5/6/8_rev 4 | reverse | | 4 |
| | DBLα1.5/6/8_rev 5 | reverse | | 4 |
| DBLα2/1.1/2/4/7/9 | DBLα2/1.1/2/4/7/9_ fwd 1 | forward | 60 | 10 |
| | DBLα2/1.1/2/4/7/9_ fwd 2 | forward | | 10 |
| | DBLα2/1.1/2/4/7/9_ rev 1 | reverse | | 10 |
| | DBLα2/1.1/2/4/7/9_ rev 2 | reverse | | 10 |
| CIDRα3.1/2 | CIDRα3.1/2_fwd 1 | forward | 60 | 7 |
| | CIDRα3.1/2_fwd 2 | forward | | 7 |
| | CIDRα3.1/2_fwd 3 | forward | | 7 |
| | CIDRα3.1/2_rev 1 | reverse | | 20 |
| Var2csa | Var2csa_fwd | forward | 60 | 20 |
| | Var2csa_rev | reverse | | 20 |

## Total *var* expression by varATS and RUF6 qPCR

Ruf6 and varATS were quantified relative to the housekeeping gene P61. All three genes were quantified on the same day without re-freezing cDNA. P61 qPCR was performed as described above. varATS qPCR was performed using the protocol outlined in [61], using a total reaction volume of 20 µl and 2 µl of cDNA as input. Ruf6 qPCR was performed in 20 µl reaction volume using Power SYBRgreen Master Mix with 2 µl cDNA and the primers Ruf6 A fwd, Ruf6 A rev, Ruf6 B fwd, Ruf6 B rev at a final concentration of 117 nM each. Cycling conditions for Ruf6 qPCR were 10 min at 95°C followed by 40 cycles of 15 sec at 95°C, 20 sec at 54°C, 7 sec at 56.5°C, 7 sec at 59°C and 20 sec at 62°C followed by a melt curve. All qPCRs were run on a QTower3.

## AMA1-ELISA

Plasma antibody levels to Ama1 were measured as previously described [95]. Samples were measured in duplicates; quantification was performed relative to a standard curve comprised of wells coated with 250 – 0.244 ng human IgG (biotechne, human IgG Control). Detection was performed using goat-anti Human IgG coupled with alkaline phosphatase (Invitrogen) and 1-Step PNPP solution (Thermo Scientific). All samples were measured at a 1:100 dilution, if samples were outside the dynamic range of the assay, measurements were repeated at higher dilutions (1:200, 1:500, 1:1000) and the average of all dilutions falling in the assay's dynamic range used. Four plasmas of malaria naïve German donors were used as negative controls. Readout was performed using a SpectraMax 190 plate reader (Molecular Devices) at 405 nm wavelength.

## Surface recognition assay

For surface recognition assay, plasmas were heat inactivated for 30 min at 56°C and pre-depleted with uninfected RBCs. FCR3 parasites were panned using HDMEC cells as previously described [96] and used within a week and mainly

expressing IT4var16. Plasmas were incubated at a 1:10 dilution in PBS with FCR3 parasites at 4% Haematocrit, ~2% parasitaemia overnight. Then, parasites were washed three times in PBS and stained with 1:50 anti-human IgG APC (Biolegend) and 1:2000 SYBR green (Invitrogen) in PBS/2%FCS for 30 min at room temperature. After 3x washes in PBS, samples were read in the FITC and APC channels of a flow cytometer (LSR II, BD). A hyperimmune pool and plasmas from two malaria naïve donors were included as positive and negative controls. Analysis was performed using the FlowJo software.

## PfEMP1 domain antibody reactivity

Recognition of PfEMP1 domains by plasma antibodies was measured as described in [64]. Briefly, plasma samples were spun down and incubated at 1:40 dilution with PfEMP1 coupled beads for 30 min, followed by 30 min incubation with 1:3000 human secondary F(ab′)2 Goat Anti-Human IgG (Jackson ImmunoResearch) and fluorescence measurement on a Bio-Plex 2000 system. Recognition of individual PfEMP1 domains was defined as fluorescence intensity greater than the average of 6 German malaria-naïve control plasmas plus two standard deviations.

## SMART-Seq single-cell RNA-seq

Glycerolyte-frozen RBC pellets of one persisting dry season sample collected at scheduled cross-sectional timepoint in July 2020 and one clinical malaria case in the 2020 wet season were thawed and stained in 1:2000 SYBR green (Invitrogen), 1:25 APC anti-human CD71, clone OKT9 (Invitrogen,), and 1:50 APC anti-human Lineage Cocktail (BioLegend) in PBS for 30 min at 37 °C, followed by 3x washes in PBS. Part of the RBC pellet of the clinical malaria case were *in vitro* cultured at 1–2% haematocrit in RPMI 1640 (Gibco) complete medium (with L-glutamine and HEPES), 7.4% sodium bicarbonate (Gibco), 100 µM hypoxanthine (C.C.Pro) and 25 mg ml $-1$ gentamycin (Gibco) added with 0.25% Albumax II (Gibco) and 10% heat-inactivated O$^{RH+}$ plasma for ~16h at 37°C in a $CO_2$ incubator (HeraCell VIOS 250i, ThermoFisher) with a gas mixture of 5% $O_2$, 5% $CO_2$ and 90% $N_2$, before staining.

In a flow cytometer (Aria III, BD), FITC-positive and APC-negative iRBCs were single-cell sorted into 96-well PCR plates with lysis buffer (TakaraBio) and flash-frozen on dry-ice followed by storage at -70 °C until further processing. Sequencing libraries were prepared from the cells using the SMART-Seq Single Cell PLUS kit (TakaraBio) and according to the manufacturer's instructions, with 20–21 cycles of cDNA amplification, and 12–16 cycles of library amplification. Quality and quantity of cDNA and libraries were assessed using the Qubit 4.0 1X dsDNA high sensitivity assay (ThermoFisher) and the Bioanalyzer high sensitivity DNA kit (Agilent). The libraries were sequenced on an Illumina MiSeq with 2 × 300 paired-end cycle protocol using the MiSeq Reagent Kit v3 (600-cycle) (Illumina).

## Single-cell transcriptome variant analysis

Raw sequencing reads were trimmed for Illumina adapters using cutadapt [97] (-a AGATCGGAAGAGCACACGTCT-GAACTCCAGTCA -A AGATCGGAAGAGCGTCGTGTAGGGAAAGAGTGT -q 30 --pair-filter = any -m 100) and mapped to *P. falciparum* 3D7 (PlasmoDB v60) [98] and human (USCS hg38) [99] using STAR [100] using the '--quantMode Gene-Counts' flag to create a gene count matrix, and the '--outSAMtype BAM SortedByCoordinate' to sort the resulting bam file. Gene annotations were obtained from PlasmoDB (v60) and GENCODE (release 34) [101]. The gene count matrix were used to quality-filter the cells (n = 186), keeping cells with >100 Pf genes and >1.000 Pf reads (n = 111), and to determine the parasite developmental stage using the scmapCell function of scmap [102] and the single cell reference atlas of Dogga et al [103]. The Bam files were filtered for mappings to the *Pf* genome and analysed for nucleic acid variations using GATK (v4) [66] with the following series of commands: AddOrReplaceReadGroups (--RGPL ILLUMINA --RGPU MiSeq --RGSM $cell_id --RGLB $sample_id), MarkDuplicates, SplitNCigarReads, BaseRecalibrator (--known-sites 3d7_hb3.vcf --known-sites 7g8_gb4.vcf --known-sites hb3_dd2.vcf) [104], ApplyBQSR, HaplotypeCaller (-stand-call-conf 20 --emit-ref-confidence GVCF). The resulting vcf files were analysed using VarClust [105] with modifications in the script.

In brief, each cell was profiled by all variant and non-variant positions with a homozygous call and coverage of at least 5 reads. The profiles were pairwise compared to generate a distance matrix based on matching variant or non-variant calls relative to the overall overlap of positions (1 – matches/overlap). All pairwise comparisons had > 5.000 overlapping positions. The varclust_heatmap function was used to generate a heatmap with hierarchical clustering based on the distance matrix.

### Single-cell *var* gene expression analysis

Trimmed reads of quality-filtered cells (n = 111) were mapped to the varDB also used for Varia (https://github.com/Thomas-DOtto/varDB/tree/master/Datasets/Varia). FeatureCount [106] (-p --primary -M -O -f) was used to generate a *var* domain count matrix, and the cells were filtered by at least 3 mapped reads (n = 90) and coverage of more than just ATS domains (n = 89). In R, the count matrix was normalized by relative counts and a scaling factor of 10.000, and the *var* domains were filtered by >100 summed normalized reads across cells in a sample. Then, a distance matrix was generated using binary metrics and used for hierarchical clustering using ward.D2 method [107]. The cells were split into 8 (dry season sample) or 24 (clinical malaria case) clusters, and clusters with similar coverage of NTS, DBLα and/or CIDRα domains were manually collapsed to a total of 7 and 22 clusters, respectively. The covered regions of similar domains were tested for sequence similarity by megablast [108] (https://blast.ncbi.nlm.nih.gov/Blast.cgi?PROGRAM=blastn).

## Supporting information

**S1 Fig. DBLα-tag sequencing in clinical and asymptomatic infections from across the year.** (**A**) Correlation of parasitaemia and participant age in October 2019 in 91 samples from beginning of the dry season (startDry n = 30), end of the dry season (endDry n = 28) and clinical malaria cases (MAL n = 24) and the wet season (midWet n = 22). (**B**) Correlation of complexity of infection (COI) and participant age in 79 (startDry (n = 20), endDry (n = 23) and midWet (n = 15)) asymptomatic infections and clinical malaria cases (MAL n = 21). (**C**) Correlation between DBLα-tag abundance between clusters in 60 samples with duplicates of > 500 reads. (**D**) Read count in 82 samples (startDry = 17, endDry n = 24, MAL n = 24, midWet n = 17) from the four timepoint after quality filtering. (**E**) Correlation between read count and number of DBLα-tags in a sample with color indicating timepoint. (F) Correlation between number of DBLα-tag clusters in the same samples and participant age with color indicating timepoint.
(TIF)

**S2 Fig. DBLα-tag sequencing in clinical and asymptomatic infections after short term culture.** Comparison of DBLα-tags in 16 wet season asymptomatic (midWet) samples (left) and 20 MAL (right) ex vivo samples (0h) and the corresponding short-term culture timepoints (12h). Individual clusters are plotted on the y-axis with lines connecting clusters detected at the 0h and 12h timepoint. Dot size indicates proportion of reads in sample mapping to the cluster. DBLα-tags were annotated with DBLα subtype by best BLAST hit in a database of annotated *var* genes.
(TIF)

**S3 Fig. DBLα-tag cluster distribution in clinical and asymptomatic infections.** (**A**) Minimum number of DBLα-tags in 65% of reads in 82 samples from clinical malaria cases and asymptomatic infections (startDry = 17, endDry n = 24, MAL n = 24, midWet n = 17). Median and IQR are shown with dot size indicating participant age. Kruskall-Wallis test with Bonferroni multiple comparison correction (**B**) Simpson diversity index of read distribution in the same 82 samples. Median and IQR are shown with dot size indicating participant age. Kruskall-Wallis test with Bonferroni multiple comparison correction. (**C**) Proportion of DBLα-tags annotated to DBLα-subtypes. Annotation is based on the best Blast hit in an annotated *var* database (min e-value 1e-02). Shown is the proportion of clusters detected at each of the timepoints belonging to the three DBLα-subtypes. (**D**) Proportion of DBLα-tag sequencing reads mapping to DBLα-tag clusters annotated as DBLα0 (left panel), DBLα1 (middle panel) or DBLα2 (right panel) in 82 samples across the four timepoints (startDry = 17, endDry

n = 24, MAL n = 24, midWet n = 17). Annotation as described above, median and IQR are indicated on the plot with dot size showing participant age. Kruskall-Wallis test with Bonferroni multiple comparison correction.
(TIF)

**S4 Fig. AMA1 amplicon sequencing and DBLα-tag sequencing in paired samples from Jan and May. (A)** AMA1 haplotype sharing between paired samples of 8 study participant at the beginning (startDry) and end of the dry season (endDry) 2019. Each box corresponds to a study participant, participant ID is shown in the gray shaded area. Dots represent a haplotype, identifiers are listed on the y-axis, with dot size indicating cluster abundance as proportion of reads. Clusters present at both timepoints are connected by a line. Two replicates per sample, haplotypes were clustered by 0.989 similarity and had to be present in both replicates. Red boxes indicate samples were all haplotypes present at the end of the dry season are also detected at the season's start. **(B)** DBLα-tag sharing between paired samples of 9 study participants at start and end of the 2019 dry season, each box corresponds to a study participant with ID shown in the gray shaded area. Dots represent a DBLα-tag, shared tags between timepoints are connected by a line. Color indicates annotation by best BLAST hit (min value 1e-02) in an annotated *var* database, abundance as proportion of reads in the sample is shown as dot size. Samples of study participants are shown in the same position in A and B panels.
(TIF)

**S5 Fig. *Var*-gene types across timepoints. (A)** Expression of putative ICAM1-binding *var* genes in 82 samples (startDry = 17, endDry n = 24, MAL n = 24, midWet n = 17). *var* genes with predicted CIDRα1 plus DBLβ1/3, were assigned to group A and CIDRα2 – 6 plus DBLβ5 domains to group B ICAM-1-binders. **(B)** Expression level of DBLδ and DBLγ containing *var* genes in the same samples. **(C)** Sankey plot of comparing annotation of DBLα-tags with cUPS and upsAI. **(D)** Relative expression of different *var*-gene domain types quantified in 54 samples (startDry n = 15, endDry n = 24, MAL n = 15, midWet n = 10) by qPCR normalized to a housekeeping gene (Fructose-bisphosphate aldolase, PF3D7_1444800). **(E)** Correlation of relative expression of different *var*-gene domain types measured by RT-qPCR in 54 samples (startDry n = 15, endDry n = 24, MAL n = 15, midWet = 10) and proportion of reads in DBLα-tag clusters belonging to *var*-genes with the equivalent domain types based on Varia prediction in the same samples. Color indicates timepoint. **(F)** var2csa expression in clinical and asymptomatic infections (startDry n = 4, endDry n = 11, MAL n = 15, Oct n = 10) and clinical malaria cases after short time of in vitro culture (MAL 12h, n = 15). B, C, E and G Kruskall-Wallis test with Bonferroni multiple comparison correction.
(TIF)

**S6 Fig. DBLα-tag sequencing and immunity. (A)** Gating strategy of surface recognition assay (SRA) with FCR3 iRBC incubated with plasma of pooled plasma of Malian individuals and stained with SYBRgreen, first gated on single cells based on forward and side scatter (top two small plots each panel) and then on infected RBC based on SYBRgreen staining in the FITC channel (bottom small plot each panel). Surface recognition was identified by staining with an anti-human IgG APC antibody and detected in the APC channel (big plot each panel). **(B)** Correlation of humoral immunity measured by SRA (left) and AMA1 ELISA (right) in plasmas from dry and wet season (startDry 19 n = 17, endDry n = 10, midWet n = 17, MAL n = 10) to participant age with color indicating timepoint. **(C)** Correlation of humoral immunity measured by and AMA1 ELISA (top) and SRA (bottom)in the same plasmas and breadth of recognition of var domains measured in Luminex. **(D)** Correlation between the different measures of humoral immunity and participant age. **(E)** Correlation of humoral immunity measured by AMA1 ELISA (left) and SRA (right) and parasitaemia.
(TIF)

**S7 Fig. Sequence comparison of *var* mapping reads in smart-seq data. (A)** Comparison of regions with sequencing coverage by smart-seq of an end-Dry season asymptomatic infection. Cells were clustered by *var* expression into 7 clusters indicated by color, and coverage of individual *var* domains in a reference database shown by cluster. The y-axis

delineates identifiers of the individual domains with mapping reads, the x-axis shows sequencing coverage by region of the respective domains. Number of cells with coverage of a specific domain type is shown next to the sequence identifier. 11 instances of reads mapping to different reference domains of the same domain type in the same cluster were detected, highlighted by brackets on the plot. Overlapping regions with coverage in the same cluster (indicated by arrows) were compared by BLAST, sequence similarity of sequences in percent is shown on the right of the respective sequence. In cluster 1, we observed mapping to different regions of different sequences of the same domain type (indicated by dashed line). In cluster 2 and 4, mapping across the majority of cells was congruent, only a single cell showed a divergent expression pattern. (**B**) Proposed *var* gene domain architecture of clusters obtained by single-cell RNAseq of end-dry season sample. Top panel shows domain composition of var genes overall, below reference domains detected by single-cell RNA-seq are listed in the corresponding positions for each cluster. (**C**) Equivalent plot to A describing *var* read mapping in a clinical malaria case.
(TIF)

**S1 Table. Expression and annotation of individual DBLα-tags in 82 clinical and asymptomatic infections.** DBLα-tag sequences were clustered by 0.96 similarity and expression level in individual samples determined. DBLα-tags were annotated using the Varia [56] and cUPS [57] tools, and by the best BLAST hit in an annotated var gene database [18]. COI, measured by ama1 amplicon sequencing and parasitaemia by varATS qPCR are also shown in the table.
(XLSX)

**S2 Table. *var* gene expression in single cells from an asymptomatic infection at the end of the dry season by SMARTSeq.** Only cells with more than three var mapping reads were included. Expression of *var* domains included in a database of 200.000 *var*s and one million *var* domains [19] in the 48 iRBCs collected at the end pf the dry season.
(XLSX)

**S3 Table. *var* gene expression in single cells from a clinical malaria case in the wet season by SMARTSeq.** Only cells with more than three var mapping reads were included. Expression of var domains included in a database of 200.000 vars and one million *var* domains [19] in the 41 iRBCs collected during a clinical malaria case in the wet season.
(XLSX)

**S4 Table. Key resources table.**
(DOCX)

## Acknowledgments

We thank the residents of Kalifabougou, Mali, for their participation in the study. Recombinant AMA1 protein was kindly provided by Karthik Gunalan and Louis H Miller, NIH. Katrin Lehmann and Mir-Farzin Mashreghi at DRFZ assisted the sequencing of the SMART-seq libraries on the MiSeq. We thank Aaron Pangilinan and Thomas Otto (University of Glasgow) for sharing upsAI tool prior to its publication. We acknowledge the use of PlasmoDB (https://plasmodb.org) throughout this study and thank its team for maintaining this valuable resource.

## Author contributions

**Conceptualization:** Sukai Ceesay, Martin Kampmann, Lasse Votborg-Novél, Thomas Lavstsen, Silvia Portugal.

**Data curation:** Sukai Ceesay, Martin Kampmann, Lasse Votborg-Novél, Helle Smedegaard Hansson, Rasmus Weisel Jensen.

**Formal analysis:** Sukai Ceesay, Martin Kampmann, Lasse Votborg-Novél, Silvia Portugal.

**Funding acquisition:** Peter D Crompton, Thomas Lavstsen, Silvia Portugal.

**Investigation:** Sukai Ceesay, Martin Kampmann, Lasse Votborg-Novél, Helle Smedegaard Hansson, Rasmus Weisel Jensen, Manuela Carrasquilla, Hamidou Cisse, Louise Turner, Usama Dabbas, Christina Ntalla, Silke Bandermann, Aissata Ongoiba, Silvia Portugal.

**Methodology:** Sukai Ceesay, Martin Kampmann, Lasse Votborg-Novél, Helle Smedegaard Hansson, Rasmus Weisel Jensen, Manuela Carrasquilla, Hamidou Cisse, Louise Turner, Usama Dabbas, Christina Ntalla, Silke Bandermann, Thomas Lavstsen, Silvia Portugal.

**Project administration:** Boubacar Traore, Peter D Crompton, Silvia Portugal.

**Resources:** Safiatou Doumbo, Didier Doumtabe, Aissata Ongoiba, Kassoum Kayentao, Boubacar Traore, Peter D Crompton, Silvia Portugal.

**Software:** Martin Kampmann, Lasse Votborg-Novél.

**Supervision:** Boubacar Traore, Peter D Crompton, Thomas Lavstsen, Silvia Portugal.

**Validation:** Martin Kampmann.

**Visualization:** Sukai Ceesay, Martin Kampmann, Lasse Votborg-Novél.

**Writing – original draft:** Sukai Ceesay, Martin Kampmann, Lasse Votborg-Novél, Thomas Lavstsen, Silvia Portugal.

**Writing – review & editing:** Sukai Ceesay, Martin Kampmann, Lasse Votborg-Novél, Helle Smedegaard Hansson, Rasmus Weisel Jensen, Manuela Carrasquilla, Hamidou Cisse, Louise Turner, Usama Dabbas, Christina Ntalla, Silke Bandermann, Safiatou Doumbo, Didier Doumtabe, Aissata Ongoiba, Kassoum Kayentao, Boubacar Traore, Peter D Crompton, Thomas Lavstsen, Silvia Portugal.

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
