## [Decision Letter · Decision Letter 0]

30 Jan 2025

PPATHOGENS-D-25-00021

Plasmodium falciparum expresses fewer var genes at lower levels during asymptomatic dry season infections than clinical malaria cases

PLOS Pathogens

Dear Dr. Portugal,

Thank you for submitting your manuscript to PLOS Pathogens. After careful consideration, we feel that it has merit but does not fully meet PLOS Pathogens publication criteria as it currently stands. Therefore, we invite you to submit a revised version of the manuscript that addresses the points raised during the review process.

Please submit your revised manuscript within 30 days Mar 31 2025 11:59PM. If you will need more time than this to complete your revisions, please reply to this message or contact the journal office at plospathogens@plos.org. Please include the following items when submitting your revised manuscript:

We look forward to receiving your revised manuscript.

Kind regards,

Ron Dzikowski

Academic Editor

PLOS Pathogens

Margaret Phillips

Section Editor

PLOS Pathogens

Sumita Bhaduri-McIntosh

Editor-in-Chief

PLOS Pathogens

orcid.org/0000-0003-2946-9497

Michael Malim

Editor-in-Chief

PLOS Pathogens

orcid.org/0000-0002-7699-2064

**Additional Editor Comments:**

As you can see all 3 reviewers (and myself included) who are experts in the field have positive opinion on this study. In your revision please address all their comments and primarily those which are listed as major points by reviewers #1. If the data is available please include it but if not please refer to it in the text as requested.

**Journal Requirements:**

At this stage, the following Authors/Authors require contributions: Sukai Ceesay, Martin Kampmann, Lasse Votborg-Novel, Helle Smedegaard Hansson, Rasmus Weisel Jensen, Manuela Carrasquilla, Hamidou Cisse, Louise Turner, Usama Dabbas, Christina Ntalla, Silke Bandermann, Safiatou Doumbo, Didier Doumtabe, Aissata Ongoiba, Kassoum Kayentao, Boubacar Traore, Peter D Crompton, Thomas Lavstsen, and Silvia Portugal. Please ensure that the full contributions of each author are acknowledged in the "Add/Edit/Remove Authors" section of our submission form.

https://journals.plos.org/plospathogens/s/submission-guidelines#loc-parts-of-a-submission

5) We notice that your supplementary Figures are included in the manuscript file. Please remove them and upload them with the file type 'Supporting Information'. Please ensure that each Supporting Information file has a legend listed in the manuscript after the references list.

**Reviewers' Comments:**

Reviewer's Responses to Questions

**Part I - Summary**

Reviewer #1: In this manuscript, the authors investigated the expression of parasite virulence genes in a Malian cohort with seasonal malaria. In particular, they sought to explain the differences in circulating parasite stages between wet and dry season parasites observed in a previous publication, suggesting differences in cytoadhesion of infected cells as the cause. In this work, they included samples of clinical cases from the wet season and samples from asymptomatic patients at the beginning and end of the dry season, but also from the middle of the rainy season as an additional control. First, the cohort is characterized in terms of parasite prevalence, parasitemia, complexity of infection, age and later the degree of immunity and these parameters are linked to the expression pattern of the var genes of the parasites, which they determined using an expressed sequence tag approach (DBLa-tag) in conjunction with previously published bioinformatics tools to predict the full-length encoded PfEMP1 protein (Varia) and its var group identity (cUPS) by its upstream region. In addition, they performed SMART scRNAseq on one isolate each from the wet and dry seasons, a method that generates reads from the full-length var genes and enables the determination of the composition of the encoded domains. In principle, methods and tools used are well established in the field, but their application to a seasonal malaria cohort is novel. From their data, they conclude that the diversity of var gene expression is limited in asymptomatic cases, with the samples at the end of the dry period showing the most extreme phenotype of very homogeneous expression of mostly a single variant and the clinical cases representing the other extreme with very diverse var gene expression. In addition, the overall level of var gene expression is reduced in asymptomatic cases. But apart from finding more EPCR-binding variants associated with severity in clinical cases, no difference in predicted binding phenotype or var group could be detected that could account for the reduced cytoadhesion.

Overall, this is a very convincing study that confirms an earlier publication by Warimwe et al. 2013 (PMID: 23922996), in which the authors described the same trend of asymptomatic infections having parasites with less diverse var expression, lower parasitemia, being older and having a higher proportion of infected erythrocyte surface-recognizing antibodies compared to non-severe and severe cases. However, as this older data set was generated by DBLa tag PCR with cloning and Sanger sequencing of a limited number of clones, the data presented is of significantly higher quality, and samples from the end of the dry season may even have a stronger “asymptomatic” phenotype. Especially, the controls included by the authors in the experimental design are very useful, such as the confirmation of DBLa-tag results with a subset of samples and genes using variant-specific qPCR primers, doing technical replicates or the exclusion of samples falling below a certain threshold for parasite RNA content to ensure comparability. The scRNA data strongly support the differences in var expression diversity between asymptomatic and clinical cases, although the overall level of var gene expression was not analyzed using this method, only a limited number of cells were analyzed and the data are likely to be difficult to grasp for non-var experts.

Reviewer #2: The submission from Ceesay and colleagues describes a detailed examination of var gene expression in individuals infected by P. falciparum in Mali. The authors analyzed samples at different points in the transmission season and in individuals that had different levels of anti-malaria immunity. The study site is particularly interesting due to the stark differences in transmission rates across the length of the year, with 6 months of intense transmission and 6 months of very little transmission. This provided the authors with the ability to investigate how var gene expression changes with the acquisition of immunity. They are also able to examine infections that they could be certain had not been recently obtained. In addition, they could compare symptomatic and asymptomatic infections. The authors obtained data indicating that as individuals acquire greater anti-PfEMP1 immunity, var expression levels fall, the number of var genes that are detectably expressed decreases, and symptoms similarly decrease. Many of these conclusions were anticipated and make sense base on current models of immunity, nonetheless validation by this study is valuable. The more provocative conclusions are that parasites might respond to anti-PfEMP1 immunity by reducing total var gene expression, something that has been hinted at in laboratory studies but not be described in the field. In addition, the authors found no correlation with PfEMP1 type and dry vs wet season transmission, which contrasts with some previous predictions. Overall, this is a valuable dataset from an interesting study that will be valuable to researchers interested in the dynamics of PfEMP1/var expression during natural malaria transmission.

Reviewer #3: The study “Plasmodium falciparum expresses fewer var genes at lower levels during asymptomatic dry season infections than clinical malaria cases” by Ceesay et al. investigated var gene expression in asymptomatic infections and clinical malaria cases among Malian children during dry and wet seasons. Sequencing and bioinformatics analyses revealed that parasites from asymptomatic infections expressed fewer var genes, with transcripts dominated by one or a few var types, while clinical cases exhibited broader var gene expression at lower levels. Parasites from asymptomatic carriers expressed a mix of CD36- and EPCR-binding PfEMP1, with CD36-binding variants dominating in non-severe cases. The findings suggested that host immunity was key in limiting var transcript diversity.

The manuscript is well written, uses state-of-the-art techniques and provides novel insights into malaria pathogenesis and persistence. It thus is of interest for a broader community. I only have some minor remarks.

**Part II – Major Issues: Key Experiments Required for Acceptance**

Reviewer #1: 1. My main concern is the claim that the overall level of var expression decreases during the dry season to sustain long-term infections. This data point is based solely on qPCR results using an ATS primer designed for diagnostic purposes, which according to the original paper (PMID: 25734259) only covers 40% of all variants and mainly B-type var genes (17 out of 19 genes in 3D7 that can be amplified from 3D7 according to Primer3 Blast). Instead of a decrease in the total amount, this could also be interpreted as a shift towards other var groups that are not covered by the primer. Especially since clinical cases are younger and less immune, I would assume that they express many B-types and may therefore be better covered by the primer. Therefore, I would be cautious to draw conclusions about the total var value solely based on this primer. Do the authors have additional data to support this point? I suggest backing this up with another approach such as the scRNAseq data. Can the authors infer anything from their scRNAseq data, e.g. more var reads in the malaria case, higher coverage of the LARSFADIG motif (as in PMID: 38270586)? Otherwise, this needs to be toned down.

2. The prediction of the upstream region and associated var gene groups from DBLa tags using cUPS is not reliable, especially the distinction between B and C types (accuracy of 48.3 %) (Preprint Tan et al 2023; doi: 10.1101/2023.11.05.565723). Thus, it remains unclear whether a specific var group signature is associated with asymptomatic infections, as this question cannot be answered with the chosen approach and requires the development of new methods. Unfortunately, this technical limitation is not discussed.

Reviewer #2: None

Reviewer #3: None

**Part III – Minor Issues: Editorial and Data Presentation Modifications**

Reviewer #1: Unfortunately, the lines in the manuscript are not numbered, which makes it somewhat difficult to refer to specific text passages.

1. Abstract: There is evidence for older stages in the circulation, which might indicate reduced cytoadhesion – as stated at the end of intro –, not vice versa as stated in the abstract: “(…) shows reduced cytoadhesion of infected erythrocytes, evidenced by the circulation of further developed parasite stages (…)”.

2. Abstract: “However, by RNAseq and qRT-PCR we also observed significantly higher total var transcript levels in malaria cases compared to asymptomatic carriers.” As far as I have seen, var levels were only assessed by RT-qPCR, not by RNAseq. Please clarify!

3. Please add reference PMID: 14651636 for var subgrouping (together with Lavstsen 2003, ref29) in the introduction.

4. Please add studies on var gene expression in asymptomatic infections in the introduction and discussion: PMID: 15122533 (even a longitudinal study) and PMID: 23922996. Please specify which studies suggest a link between asymptomatic infections and varC expression.

5. To further improve the quality of the assumptions that can be made, the duration of infection should be assessed as an additional variable. Can the authors use regularly collected dried blood spots/venous blood samples and determine the COI/genotype and assess how long the individuals were infected and also correlate this with var gene expression, parasitemia, etc. as they have done in PMID: 39284949?

6. Parasites from clinical cases that had been cultured for 12 (DBLa-tag) or 16 hours (scRNAseq) served as controls in some experiments. However, the difference in the mean age of wet and dry season parasites is only about 6 hours, as shown in a previous publication (PMID: 35927582). Please comment!

7. It has already been shown that changes in var gene expression can occur during short-term in vitro cultivation in samples, although these cannot be predicted (PMID: 38270586). This could serve as an additional argument on page 6 (section before Figure 1).

8. Figure 1F: A Blant-Altman plot should be used to compare two methods, please replace.

9. It would be helpful if the pseudotimes for the parasites analyzed by RNAseq could be provided to assess the homogeneity of the developmental age of the parasites within the asymptomatic and clinical sample.

10. Please provide details of the primers used in the study. Currently there is not even a reference in the methods section for the P61, P90 and no details at all for the qPCR primers used to validate the DBLa tag results.

11. On page 4 it is stated that RNA was assessed with two housekeeping genes to select samples with sufficient parasite RNA. However, the “Methods” section (page 22) states that only P90/seryl-tRNA was used to select samples (<ct30). at="" did="" fba="" in="" look="" p61="" qpcr="" the="" why="" you="">12. Please provide the formula for calculating the Simpson Diversity Index in the Methods section.

13. Figure 2: I would suggest showing Figure S3B instead of Figure 2C in the main figures, as the diversity index is based on both measurements, the number of clusters found and the proportions.

14. Figure 3: Since expression of the non EPCR-binding A-type (“unknown A” binding phenotype) was assessed in qPCR (DBL1.5/6/8), I would suggest including the predicted CIDRb/g/d domains in panel A as well, especially since the binding phenotype has also been associated with severe disease.

15. The overviews hidden in the supplement (Figure S3C, S5A & D) are easy to understand and should, in my opinion, be included in the main figures – especially as there are no obvious differences between the samples.

16. Recently, ruf6 expression was shown to be transcribed from PolIII and regulated by Maf1, and an increase in ruf6 and var expression was associated with increased plasma magnesium levels in symptomatic wet season samples (PMID: 38921824). Do you have any data from your cohort to support this study?

17. Page 12: Ref29 is not the correct reference for the Luminex assay

18. To improve the flow of the manuscript, I would suggest moving the section on the assessment of immunity to the first chapter on the description of the cohort, as it fits with the age and parasitemia curves and then the scRNA results are not so detached.

19. Also, is there an assessment of immunity in this cohort of those who were able to overcome the infection compared to those who remained positive throughout the dry season or were negative at the beginning of the dry season?

20. The entire section on scRNA may be difficult to understand for non-PfEMP1 experts. The mapping of the sc reads against varDB, in which the genomic sequences of the var genes of the isolates studied are not deposited, was performed because no reference genomes are available. Why did the authors not generate a reference, either by sequencing gDNA or by de novo assembly of var transcripts from RNAseq reads? At the very least, I would recommend providing schematics for each cluster with the putative domain composition, which would also show that different regions of a domain correspond to different domains in the database.

21. Figure 6B&C: What is the difference between dots that are filled and others that contain a plus? I would suggest drawing vertical lines between the clusters.

22. Please comment if you have seen var-null or multiple var-expressing cells as described in the preprint https://doi.org/10.1101/2024.03.08.584127. Have you found any other expression differences apart from the var genes?

23. The authors provide the scRNAseq data only on request, as indicated in the legends of Tables S2 and 3. The data should be included in the manuscript supplement or deposited in publicly accessible databases.

24. We already know from CHMI infections and another cohort study using the DBLa tag approach that naïve (and also more severe cases) and less immune African individuals (“non-controllers”) express a higher number of var genes, which is restricted in individuals with higher immunity (“controllers”), this might be worth discussing (PMID: 31295334; PMID: 27070311; PMID: 33908865; PMID: 34340541; doi: https://doi.org/10.1101/2023.12.27.23300577)</ct30).>

Reviewer #2: I have no major criticisms for the authors. The present a large set of data, and while alternative methods of analysis and presentation are possible, I will leave that to the authors’ discretion.

Minor, suggestions:

1. In all the Figures, the authors occasionally use yellow or light green text (or symbols) that are virtually impossible to read on a white background. They should adjust all the colors to ensure easy legibility.

2. The legend to Figure 1 is missing a description of panel E. The current legend includes descriptions of panels F and G but labels them E and F.

3. In the first sentence on page 4 the authors use the abbreviation MAL but don’t define it until later in the paragraph.

4. On page 7, the authors annotate their DBL-tag sequences to DBLα types 0, 1 and 2. Please provide the readers with a brief description of what these types are and why this is interesting.

Reviewer #3: • The reference numbering starts at [2]

• Page 4, first line: The abbreviation MAL is used for the first time, but not defined. Later in the manuscript, MAL is defined several times or sometimes the abbreviation is not used at all but the full term. MAL should be defined one time, when the it is used for the first time, but then MAL should be used consequently throughout the text.

• Page 4, center part: "The individuals’ age correlated negatively to parasitaemia (Spearman’s cor -0.52, p= 8.99e-7) (S Fig. 1A)" These values cannot be derived from the graph. There is a correlation coefficient of -0.39 in the graph.

• Page 13, center part: "iRBC surface recognition , and..." There is an extra space.

• Page 13, last line of the text: The authors talk about a negative correlation, so it should be -0.71 instead of 0.71. (It is also incorrect in the graph.)

• Page 14, center part: Reference to Fig. 6a - it should be 6A.

• Page 19, line 4: Fig.4 - a space is missing here.

• Materials and Methods section: µ is italicized in some places and not in others.

• Page 20, end of the first paragraph: One study exclusion criterion is a temperature of greater than or equal to 37.5 degrees? Does this make sense when the authors further write the following: Clinical malaria cases were defined as Plasmodium infection with parasitaemia >2500 parasites/µl blood and an axillary temperature >37.5°C?

• Page 20, last paragraph: P. falciparum is not italicized.

• Page 25, second paragraph: "Recombinant Pf3D7 AMA1." Incomplete sentence?

• Page 26, center part: flash-frozen instead of "flash-freezed."

• Figure 1: The legend lacks a description of E; the panel labeled as E should be F, and what is labeled as F should be G. In the last two sentences of the legend, the statistical tests assigned to the panels are also incorrect.

• Figure 2: The legend is missing an L in the last sentence at "individual."

• Figure S4: The legend states, "Red boxes indicate samples where all haplotypes present at the end of the dry season are also detected at the season’s start." This cannot be seen. Does this only apply to A? Then what do the boxes mean in B?

• Figure S6: I believe the two panels of E do not appear in the text and also lack a legend. Does the legend for C and D match? The legend does not fit the figure.

PLOS authors have the option to publish the peer review history of their article (what does this mean? ). If published, this will include your full peer review and any attached files.

**Do you want your identity to be public for this peer review?** For information about this choice, including consent withdrawal, please see our Privacy Policy .

Reviewer #1: No

Reviewer #2: No

Reviewer #3: No

**Figure resubmission:**
---

## [Decision Letter · Decision Letter 1]

12 May 2025

Dear Dr. Portugal,

We are pleased to inform you that your manuscript 'Plasmodium falciparum expresses fewer var genes at lower levels during asymptomatic dry season infections than clinical malaria cases' has been provisionally accepted for publication in PLOS Pathogens.

Best regards,

Ron Dzikowski

Academic Editor

PLOS Pathogens

Margaret Phillips

Section Editor

PLOS Pathogens

Sumita Bhaduri-McIntosh

Editor-in-Chief

PLOS Pathogens

orcid.org/0000-0003-2946-9497

Michael Malim

Editor-in-Chief

PLOS Pathogens

orcid.org/0000-0002-7699-2064

Reviewer Comments (if any, and for reference):

Reviewer's Responses to Questions

**Part I - Summary**

Reviewer #1: My comments have been adequately addressed.

Reviewer #2: This manuscript describes a valuable and interesting dataset that details var gene expression at different points in the transmission season and in individuals that had different levels of anti-malaria immunity. The authors provide convincing evidence that as individuals acquire greater anti-PfEMP1 immunity, var expression levels fall, the number of var genes that are detectably expressed decreases, and symptoms similarly decrease. The analysis is extensive and thorough and provides insights into the dynamics of lengthy P. falciparum infections and the acquisition of immunity.

Reviewer #3: (No Response)

**Part II – Major Issues: Key Experiments Required for Acceptance**

Reviewer #1: n/a

Reviewer #2: None.

Reviewer #3: (No Response)

**Part III – Minor Issues: Editorial and Data Presentation Modifications**

Reviewer #1: Point 2: Analyzing the DBLa tag data using the new upsAI tool with increased accuracy and increased number of predicted DBLa tags (>99%) improved the significance of this data point. However, it also shows the difficulty in distinguishing between upsB and upsC types, as these values differed the most between the two prediction tools. Unfortunately, as the upsAI tool is not yet published, it is difficult to assess whether reliable predictions of ups types can be made from relatively short DBLa tags or whether longer sequences are required. For correct interpretation of the data, I think any remaining limitations/difficulties in distinguishing between upsB and upsC with cUPS and upsAI should be mentioned somewhere in the manuscript.

Line 272: var should be in italics

Line 254: Instead of “encoded var domains” -> “var-encoded domains” or “encoded PfEMP1 domains”

Reviewer #2: The authors have suitably addressed my previous comments. I have no additional suggestions.

Reviewer #3: (No Response)

PLOS authors have the option to publish the peer review history of their article (what does this mean? ). If published, this will include your full peer review and any attached files.

**Do you want your identity to be public for this peer review?** For information about this choice, including consent withdrawal, please see our Privacy Policy .

Reviewer #1: No

Reviewer #2: No

Reviewer #3: **Yes: ** Gabriele Pradel

---

## [Editor Report · Acceptance letter]

Dear Dr. Portugal,

We are delighted to inform you that your manuscript, "Plasmodium falciparum expresses fewer var genes at lower levels during asymptomatic dry season infections than clinical malaria cases," has been formally accepted for publication in PLOS Pathogens.

Best regards,

Sumita Bhaduri-McIntosh

Editor-in-Chief

PLOS Pathogens

orcid.org/0000-0003-2946-9497

Michael Malim

Editor-in-Chief

PLOS Pathogens

orcid.org/0000-0002-7699-2064